# Catalytic role of in-situ formed C-N species for enhanced $Li_2CO_3$ decomposition

Fangli Zhang[1,2,3], Wenchao Zhang [1,4] ✉, Jodie A. Yuwono [2], David Wexler[3], Yameng Fan[3], Jinshuo Zou[2], Gemeng Liang [2], Liang Sun [2] & Zaiping Guo [2] ✉

Sluggish kinetics of the $CO_2$ reduction/evolution reactions lead to the accumulation of $Li_2CO_3$ residuals and thus possible catalyst deactivation, which hinders the long-term cycling stability of Li-$CO_2$ batteries. Apart from catalyst design, constructing a fluorinated solid-electrolyte interphase is a conventional strategy to minimize parasitic reactions and prolong cycle life. However, the catalytic effects of solid-electrolyte interphase components have been overlooked and remain unclear. Herein, we systematically regulate the compositions of solid-electrolyte interphase via tuning electrolyte solvation structures, anion coordination, and binding free energy between Li ion and anion. The cells exhibit distinct improvement in cycling performance with increasing content of C-N species in solid-electrolyte interphase layers. The enhancement originates from a catalytic effect towards accelerating the $Li_2CO_3$ formation/decomposition kinetics. Theoretical analysis reveals that C-N species provide strong adsorption sites and promote charge transfer from interface to $*CO_2^{2-}$ during discharge, and from $Li_2CO_3$ to C-N species during charge, thereby building a bidirectional fast-reacting bridge for $CO_2$ reduction/evolution reactions. This finding enables us to design a C-N rich solid-electrolyte interphase via dual-salt electrolytes, improving cycle life of Li-$CO_2$ batteries to twice that using traditional electrolytes. Our work provides an insight into interfacial design by tuning of catalytic properties towards $CO_2$ reduction/evolution reactions.

Severe global warming has resulted in pledges by current society to achieve the goals of carbon neutrality with projections of atmospheric carbon peaking followed by eventual falls[1–3]. In this context, how to achieve eco-efficient and environmentally sustainable $CO_2$ capture, and to reconstruct energy device systems for balancing reduction of carbon emissions and growing energy demands has become a worldwide challenge[4,5]. Rechargeable Li–$CO_2$ batteries are considered potential candidates for advanced energy storage devices and $CO_2$ fixation. There is potential to achieve a win-win situation due to their relatively high discharge potential (2.80 V vs. $Li^+/Li$) and theoretical

specific energy density (1876 Wh $kg^{-1}$), based on the reversible redox reaction of $3CO_2 + 4Li^+ + 4e^- \leftrightarrow 2Li_2CO_3 + C$[6,7]. In the field of space exploration, it has been projected by NASA that uptake of Li-$CO_2$ batteries would enable significant weight and cost savings for Mars exploration missions because 96% of the Martian atmosphere is $CO_2$[8,9]. In both these areas, Li-$CO_2$ batteries represent potentially attractive options.

The sluggish kinetics of $CO_2$ reduction/evolution reactions (CRR/CER) and the poor charge transfer capability of Li-$CO_2$ batteries generally lead to the accumulation of $Li_2CO_3$ residuals and high charge

[1]School of Metallurgy and Environment, Central South University, Changsha 410083, China. [2]School of Chemical Engineering, The University of Adelaide, Adelaide, SA 5005, Australia. [3]Institute for Superconducting & Electronic Materials, University of Wollongong, Faculty of Engineering and Information Science, Wollongong, NSW 2500, Australia. [4]Chinese National Engineering Research Centre for Control & Treatment of Heavy Metal Pollution, Changsha 410083, China. ✉e-mail: wenchao.zhang@csu.edu.cn; zaiping.guo@adelaide.edu.au

potential, which further results in fast electrolyte degradation and consequent deterioration in battery cyclability[10,11]. Electrolyte engineering has been regarded as an effective and practical approach to addressing limited battery performance[12–14]. In the past, tremendous efforts have been dedicated to constructing robust inorganic-rich solid-electrolyte interphase (SEI) layers on electrodes, especially fluorinated SEIs, to suppress parasitic reactions between electrodes and electrolytes and improve battery cyclability[15–19]. Indeed, the organic components in the SEI layer have been considered to possess porous structures, which leads to an increase in the SEI thickness and exacerbates the non-uniform diffusion of Li⁺ ions at the interface[16,20]. However, detailed reaction mechanisms of the organic SEI components (e.g., C-N, C-F, C-S, etc.) and their catalytic effects have barely been studied in the past few decades, particularly in relationship to $CO_2$ reduction/evolution chemistry. Furthermore, how to achieve a fundamental understanding of the key roles of organic SEI components in determining reaction kinetics and reversibility of $Li_2CO_3$, and, finally, governing battery performance has remained a mystery for decades. Therefore, it is of great importance to initiate investigation of the organic components in SEIs with the hope of revealing their structures and corresponding features, including their formation mechanism, adsorption of the reactant gas $CO_2$, intermediate $*CO_2^{2-}$ radicals, and the discharge product $Li_2CO_3$, and corresponding catalytic activity.

Herein, we have initiated fundamental studies of the organic components in SEI layers and attempted to decipher the mysteries associated with their electrochemical performance in Li-$CO_2$ batteries. To better explore and detect the catalytic effects of SEI components, reduced graphene oxide (rGO) with poor catalytic performance was employed as a matrix on cathodes. Among the several traditional single-salt-based electrolytes, even with lower content of LiF in the SEI layer compared to that in lithium bis(trifluoromethanesulfonyl)imide (LiTFSI)-based electrolyte, the cells in lithium bis(fluorosulfonyl)imide (LiFSI)-based electrolyte containing a higher content of C-N species in the SEI layer could deliver relatively longer cycle life. Characterizations of cycled cathodes confirm that the content of C-N species is tightly

correlated with reversibility of $Li_2CO_3$, which is the key to determining the battery cycling. First-principles calculation results show that C-N species at the interface can provide strong adsorption sites and fast charge transfer capability to $*CO_2^{2-}$ and $Li_2CO_3$, which can enhance the interfacial catalytic properties to accelerate the CRR/CER kinetics. In the CRR, spontaneous charge transfer can be achieved between C-N species and $*CO_2^{2-}$, which could drive the $*CO_2^{2-}$ to quickly react with $C_2O_4^{2-}$ to generate $CO_3^{2-}$, thus facilitating the formation of $Li_2CO_3$. In the CER, the strong interaction between Li atoms in $Li_2CO_3$ and the interfacial N sites of C-N species enable fast charge transfer from $Li_2CO_3$ to C-N species, which can notably weaken Li-O bonds in $Li_2CO_3$ and decrease its decomposition energy barriers. When paired with a ruthenium (Ru) catalyst, the Li-$CO_2$ cells also follow the same tendency as above, indicating that the positive effects of C-N species towards battery cycling are applicable with the participation of a metal catalyst. To further improve battery cycling by increasing the content of C-N species, a C-N rich SEI layer was constructed in dual-salt ($LiNO_3$/ LiFSI)-based electrolytes. With this SEI layer, cells in 0.25 M $LiNO_3$/0.75 M LiFSI electrolytes could achieve stable long-term cycling of >1500 h with a metal-free catalyst of rGO and prolonged cycling of > 2200 h with an Ru catalyst. Our finding opens an avenue to address the bottleneck issue for Li-$CO_2$ batteries and provides an electrolyte design principle for guiding future electrolyte design in batteries.

## Results

### Modelling for adsorption energy and charge transfer analyses

The $CO_2$ reduction/evolution reactions are relatively complicated processes in Li-$CO_2$ batteries, occurring through multiple intermediate steps, as shown in Equations S(1)-(4) and Supplementary Fig. 1 in the Supplementary Information[21,22]. On the one hand, $*CO_2^{2-}$, as a key intermediate radical, is a crucial driving force that governs spontaneous charge transfer capability and reduces the formation energy barrier of $Li_2CO_3$, both of which dominate the $CO_2$ reaction kinetics (Fig. 1a). On the other hand, the lifespan of Li-$CO_2$ batteries is determined by the reversibility of the discharge product $Li_2CO_3$, due to its insulating nature[23]. To unveil the relationship between the $CO_2$

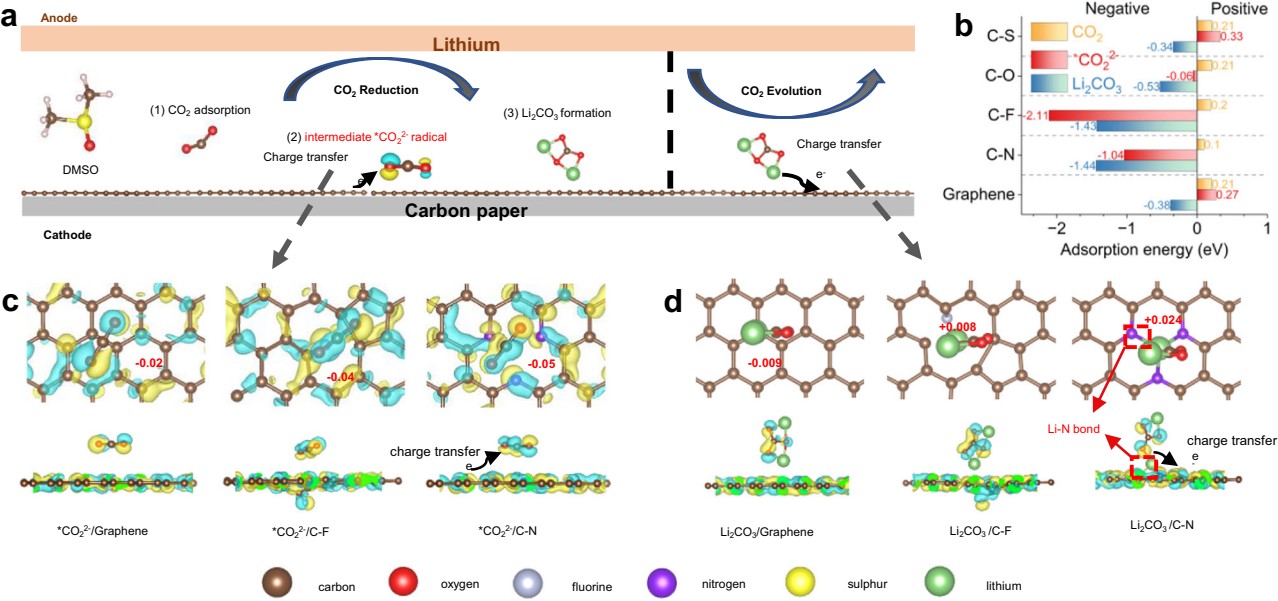

**Fig. 1 | Modelling for adsorption energy and charge transfer analyses of various SEI compositions. a** Schematic representation of the $CO_2$ reduction/evolution processes on the cathode surfaces. **b** Adsorption energies of graphene, C-N, C-F, C-O, and C-S species for $CO_2$, $*CO_2^{2-}$, and $Li_2CO_3$. Top views and side views of the atomic structures and the charge density differences of (**c**) $*CO_2^{2-}$ and (**d**) $Li_2CO_3$ adsorption on graphene, C-F, and C-N species. The red numbers are the Bader

charge values of these different species for $*CO_2^{2-}$ and $Li_2CO_3$ molecules. The yellow and blue zones represent electron loss and gain, respectively (isovalue, $2 \times 10^{-6}$). The carbon, oxygen, fluorine, nitrogen, sulphur, and lithium atoms are marked as copper, red, silver, purple, yellow, and light green, respectively.

reaction kinetics and the cathode surface chemistry, we first constructed bonding configurations for basic graphene and common organic SEI components (Supplementary Fig. 2) and compared their adsorption energy for $CO_2$, $*CO_2^{2-}$, and $Li_2CO_3$ by first-principles calculations to systematically investigate their configurations and catalytic properties (Supplementary Table 1).

In the CRR, C-N species show relatively higher adsorption energies towards $CO_2$ (0.1 eV) and $*CO_2^{2-}$ (−1.04 eV) than C-O, C-S, and graphene (Fig. 1b, yellow and red bars, respectively), indicating a lower formation barrier of $Li_2CO_3$ and higher catalytic activation on C-N species. After adsorption, the charge transfer capabilities determined by the interaction between $*CO_2^{2-}$ and the electrode surface layer are critical to governing the reaction kinetics. Therefore, as typical examples, Bader charge and charge density difference analyses for graphene, C-N, and C-F species were conducted to comprehensively study their charge transfer capability. As presented in Fig. 1c, the value of spontaneous charge transfer is the largest from C-N species to $*CO_2^{2-}$ (−0.05 e⁻), whereas its value on graphene and C-F species is smaller (around −0.02 and −0.04 e⁻, respectively). As a result, C-N species have a positive effect towards facilitating the nucleation and uniform distribution of $Li_2CO_3$ by providing strong adsorption sites and enhancing the CRR kinetics.

In the CER, the adsorption energies show that C-N and C-F species exhibit strong interactions (−1.44 and −1.43 eV, respectively) with $Li_2CO_3$, whereas graphene, C-O, and C-S species show weak interactions (Fig. 1b, blue bar). In addition, there is an evident charge transfer of 0.024 e⁻ from $Li_2CO_3$ to C-N species, which is significantly larger than that to graphene and C-F species (Fig. 1d), indicating that discharge products $Li_2CO_3$ on C-N species more easily lose electrons. It should be noted that the atomic configurations clearly show that there is a strong interaction between $Li_2CO_3$ and C-N species, revealing the weaker nature of the Li-O bonds and reduced decomposition energy barriers of $Li_2CO_3$[24,25]. As a result, it is expected that C-N species can construct a bidirectional fast-electron migration bridge by providing strong adsorption sites and fast charge transfer capability to $*CO_2^{2-}$ and $Li_2CO_3$, achieving boosted CRR/CER kinetics and enhanced interfacial catalytic properties.

## Performance evaluation of single-salt electrolytes for Li-$CO_2$ cells

We proceeded to Li-$CO_2$ cells to verify the practicality of the above theoretical results. Various 1 M lithium salts (lithium nitrate ($LiNO_3$), or LiFSI, or lithium tetrafluoroborate ($LiBF_4$), or LiTFSI), in dimethyl sulfoxide (DMSO) solvent, were selected as electrolytes to manipulate the SEI composition on cathode surfaces. To better explore the key roles of the SEI components and detect them, commercial rGO with poor catalytic performance was employed as a matrix.

Figure 2 shows the overall electrochemical performance of the Li-$CO_2$ cells, with the cells using LiFSI electrolytes (denoted as LiFSI cell) delivering superior cycling and rate capabilities among the various electrolytes. Cycling performance was evaluated by limiting the cut-off specific capacity to 500 mA h g⁻¹. The long-term profiles show that the LiFSI cell notably delivers the longest lifetime of 90 cycles (900 h) at 0.1 A g⁻¹, which was twice that in the cell using $LiNO_3$ electrolytes, denoted as $LiNO_3$ cell, delivering 45 cycles (450 h) (Fig. 2a top and Supplementary Fig. 3). When the current density was increased to 0.2 A g⁻¹, the LiFSI cell could still run for 91 cycles compared to the other cells, in which the cycling performance was degraded (Fig. 2a bottom and Supplementary Fig. 4). This clearly demonstrates the substantial effects of different salts on battery cycling. The superior cycling performance of LiFSI cells is attributed to relatively high ionic conductivity of LiFSI electrolytes (10.93 mS cm⁻¹, Supplementary Fig. 5) and excellent reversibility of $Li_2CO_3$ during cycling in LiFSI cells, as supported by electrochemical impedance spectroscopy (EIS)[8,23]. Full recovery of the impedance spectrum could only be observed for the

LiFSI cell (Fig. 2b) after a discharge-charge cycle compared with a partial recovery for the others (Supplementary Fig. 6), suggesting the complete decomposition of deposited $Li_2CO_3$ upon recharging in LiFSI cells. Furthermore, Supplementary Fig. 7 shows that the increase in impedance with cycling follows the trend of LiFSI <LiTFSI <$LiBF_4$< $LiNO_3$, consistent with their cycling profiles (cycle life decreases in the order of LiFSI > LiTFSI > $LiBF_4$ > $LiNO_3$). As expected, the impedances of LiFSI cells were maintained to be the lowest (Supplementary Fig. 7d), while those of $LiNO_3$ cells vastly increased just after 20 cycles (Supplementary Fig. 7a).

The excellent electrochemical performance of LiFSI cells was also revealed through full-discharge capability and rate performance. Figure 2c presents the full-discharge curves at 0.1 A g⁻¹, demonstrating that the LiFSI cell delivers the highest discharge capacity of 8,629.26 mA h g⁻¹ compared to the others. Rate performance was evaluated at different current densities, from 0.1 to 2 A g⁻¹. At low current densities, the charge/discharge curves of the Li-$CO_2$ cells show similar overpotential. At a high current density of 2 A g⁻¹, however, a flat discharge voltage plateau (Fig. 2d) with a small overpotential (Fig. 2e) could only be found in the LiFSI cell. The cells with other electrolytes suffered from a dramatic decay of their discharge capacity at 2 A g⁻¹ (Supplementary Figs. 8 and 9). To further test the effect of various electrolytes on battery performance, Li-$CO_2$ cells were also assembled by using carbon papers as matrix without catalysts (Supplementary Fig. 10) or using cathodes with Ru catalysts (Supplementary Fig. 11), respectively, which exhibited the same tendency of cycling performance as that using rGO as matrix. By using Ru as catalysts, the cells in LiFSI-based electrolytes maintain the cut-off capacity after 630 h at 0.3 A g⁻¹, whereas the cells in $LiNO_3$-based electrolytes exhibit capacity decay after 250 h (Supplementary Fig. 11). The superior cycling stability of Li-$CO_2$ batteries with the LiFSI electrolyte may result from high reversibility of $Li_2CO_3$ in LiFSI cells, which enables a relatively low charge voltage for CER and thus suppresses electrolyte decomposition.

## Cathode characterization to understand the correlation between reversibility of $Li_2CO_3$ and SEI composition

In-situ differential electrochemical mass spectrometry (DEMS) was first conducted to examine the reversibility of $Li_2CO_3$ in real time by monitoring $CO_2$ consumption, evolution, and the corresponding charge-to-mass ratios ($e^-$/$CO_2$). The collected gases in the in-situ DEMS results show that $CO_2$ was consumed in first battery discharge, and $CO_2$ was released in first battery recharge (Fig. 3a, b). The theoretical value of $e^-$/$CO_2$ ratio is 1.33 ($4e^-$/$3CO_2$), corresponding to the decomposition of $Li_2CO_3$ in accordance with the redox reaction of $3CO_2 + 4Li^+ + 4e^- \leftrightarrow 2Li_2CO_3 + C$[26]. The deviation from the standard value of 1.33 indicates the occurrence of parasitic reactions. As shown in Fig. 3a, the $e^-$/$CO_2$ ratios in $LiNO_3$ cells deviated considerably from 1.33, which were determined as 1.09 and 1.72 for discharge and charge processes, respectively. In contrast, the ratios in the LiFSI cells were determined to be 1.327 and 1.5 during discharge and charge, respectively (Fig. 3b). The lower deviation from the standard $e^-$/$CO_2$ value in the LiFSI cell implies better reversibility of $Li_2CO_3$ and less electrolyte decomposition than that in the $LiNO_3$ cell, consistent with superior cycling performance of the LiFSI cell.

Furthermore, Li-$CO_2$ cells were disassembled after cycling for ex-situ cathode characterization to confirm the correlation between the reversibility of $Li_2CO_3$ and battery cycling performance. Scanning electron microscope (SEM) was performed on cathodes before (Supplementary Fig. 12) and after discharge (Supplementary Fig. 13), revealing discharge products deposited on cathode surfaces. Further detailed examination using high resolution transmission electron microscopy (TEM) confirmed this (Supplementary Fig. 14), with fast Fourier transform patterns in TEM images, containing rings correspond to the (101) and ($\bar{3}$12) lattice planes of $Li_2CO_3$, which could verify

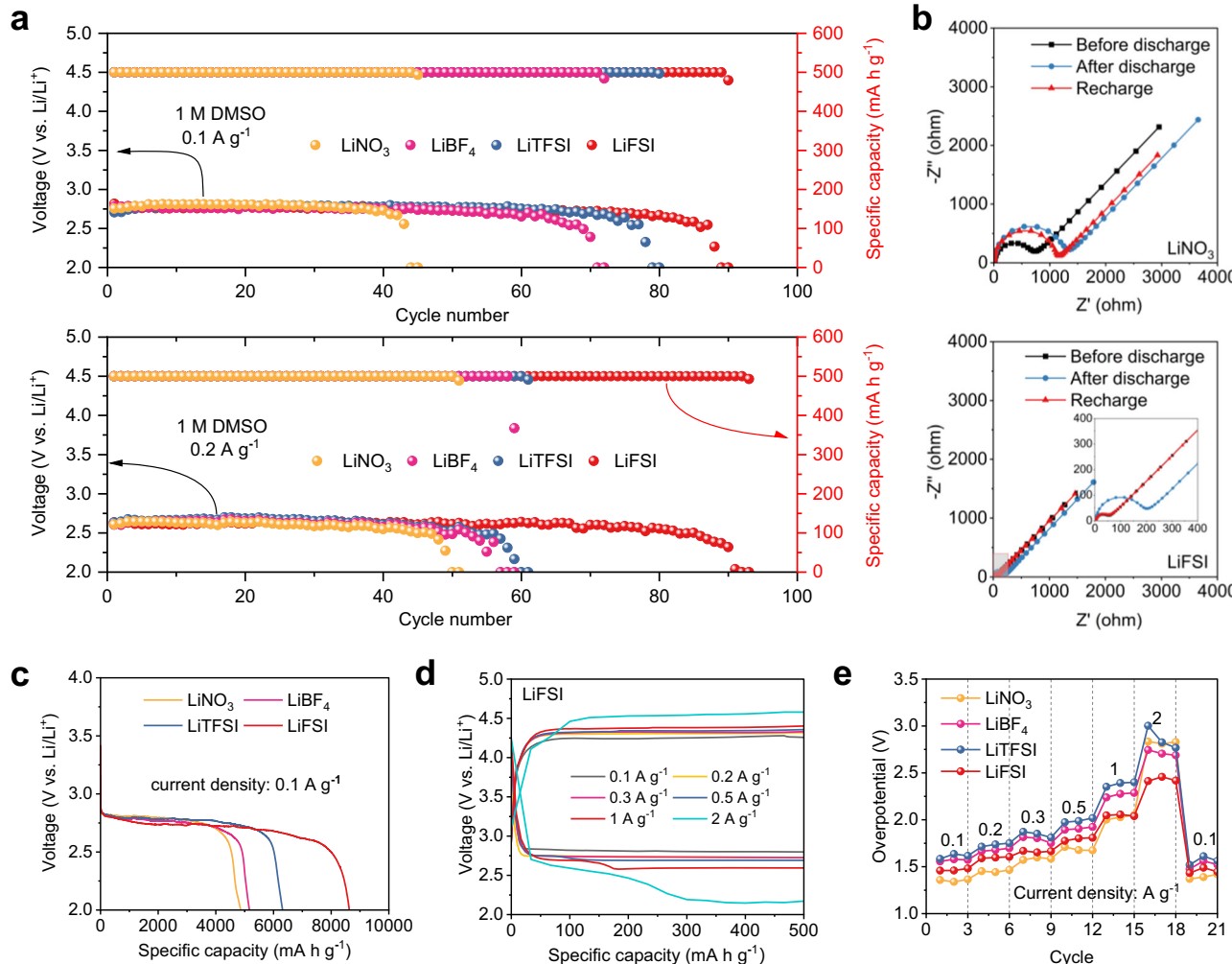

**Fig. 2 | Electrochemical performance of Li-CO₂ cells in various electrolytes.**
**a** Long-term cycling performance at 0.1 A g⁻¹ (top) and 0.2 A g⁻¹ (bottom) with a cut-off capacity of 500 mAh g⁻¹ (in the cut-off voltage from 2 V to 5 V). **b** EIS spectra of the LiNO₃ (top) and LiFSI (bottom) cells (the inset shows enlarged EIS spectra for LiFSI cells) before discharge, after the first discharge, and after recharge. **c** Full-discharge curves at 0.1 A g⁻¹ with low cut-off voltage of 2 V. **d** Discharge-charge curves of the LiFSI cell taken at variant current densities (0.1–2 A g⁻¹ in the cut-off voltage from 2 V to 5 V). **e** Overpotential comparison of the various electrolytes at different current densities.

the formation of Li₂CO₃ as discharge product. The formation of Li₂CO₃ was also confirmed by the selected area electron diffraction (SAED) pattern (Supplementary Fig. 15), the newly emerged peaks at 860 cm⁻¹, 1412 cm⁻¹ and 1473 cm⁻¹ in the Fourier transform infrared (FTIR) spectra (Supplementary Fig. 16)[27], a peak located at 1089 cm⁻¹ in the Raman spectra (Supplementary Fig. 17)[28], and a peak at 290.6 eV in the X-ray photoelectron spectroscopy (XPS) spectra (Supplementary Fig. 18)[29,30]. In order to eliminate the possible sample pollution, in-situ Raman evidence was provided for the Li-CO₂ cells to prove the formation/ decomposition of Li₂CO₃ during battery cycling (Supplementary Fig. 19). Taking LiNO₃ and LiFSI electrolytes as examples, the peak of Li₂CO₃ can be clearly observed to emerge during discharge and gradually disappear during charging (Supplementary Fig. 20), suggesting the excellent reversibility of the cathodes. Furthermore, signals of DMSO₂ species (*) were detected, which suggests decomposition of DMSO solvent due to the attack of reduced *CO₂²⁻ radicals during cycling. Compared to the strong signals of DMSO₂ species in LiNO₃ cells, the signals in LiFSI cells gradually disappeared after being fully recharged, verifying that solvent degradation could be suppressed to some extent, in accord with its superior cycling stability.

After recharge, the cathodes nearly return to the pristine morphology (Supplementary Fig. 21). More importantly, the complete Li₂CO₃ decomposition could only be observed in LiFSI cells, based on the disappearance of the Li₂CO₃ peak after cycling in FTIR (Supplementary Fig. 16) and Raman spectra (Supplementary Fig. 22), which indicates outstanding electrochemical reversibility of Li₂CO₃. In contrast, certain amounts of Li₂CO₃ residuals were still detected on the cycled cathodes in the other electrolytes. Examination of the C 1s XPS spectra further revealed the relationship between Li₂CO₃ residuals and battery cycling performance, in which the cells show the presence of Li₂CO₃ residuals except for the cell in LiFSI electrolytes (Fig. 3c left). Notably, the cells present cyclability decay as the intensity of Li₂CO₃ residuals peak becomes stronger, especially for the LiNO₃ cell, which displays the largest amount of Li₂CO₃ residuals and unsatisfactory cycling performance. These results cross-validated the importance of reversibility of Li₂CO₃ in determining cell capacity degradation and displayed consistency with their cycling profiles.

Based on our calculated results (Fig. 1), C-N species in SEIs can enhance interfacial catalytic properties to accelerate the CO₂ reaction kinetics, suggesting a positive correlation between C-N species and reversibility of Li₂CO₃. Therefore, SEI with a high content of C-N species ensures the complete decomposition of Li₂CO₃, thus resulting in excellent cycling capability. Ex-situ XPS analysis of cathodes before (Supplementary Fig. 23) and after cycling (Supplementary Fig. 24)

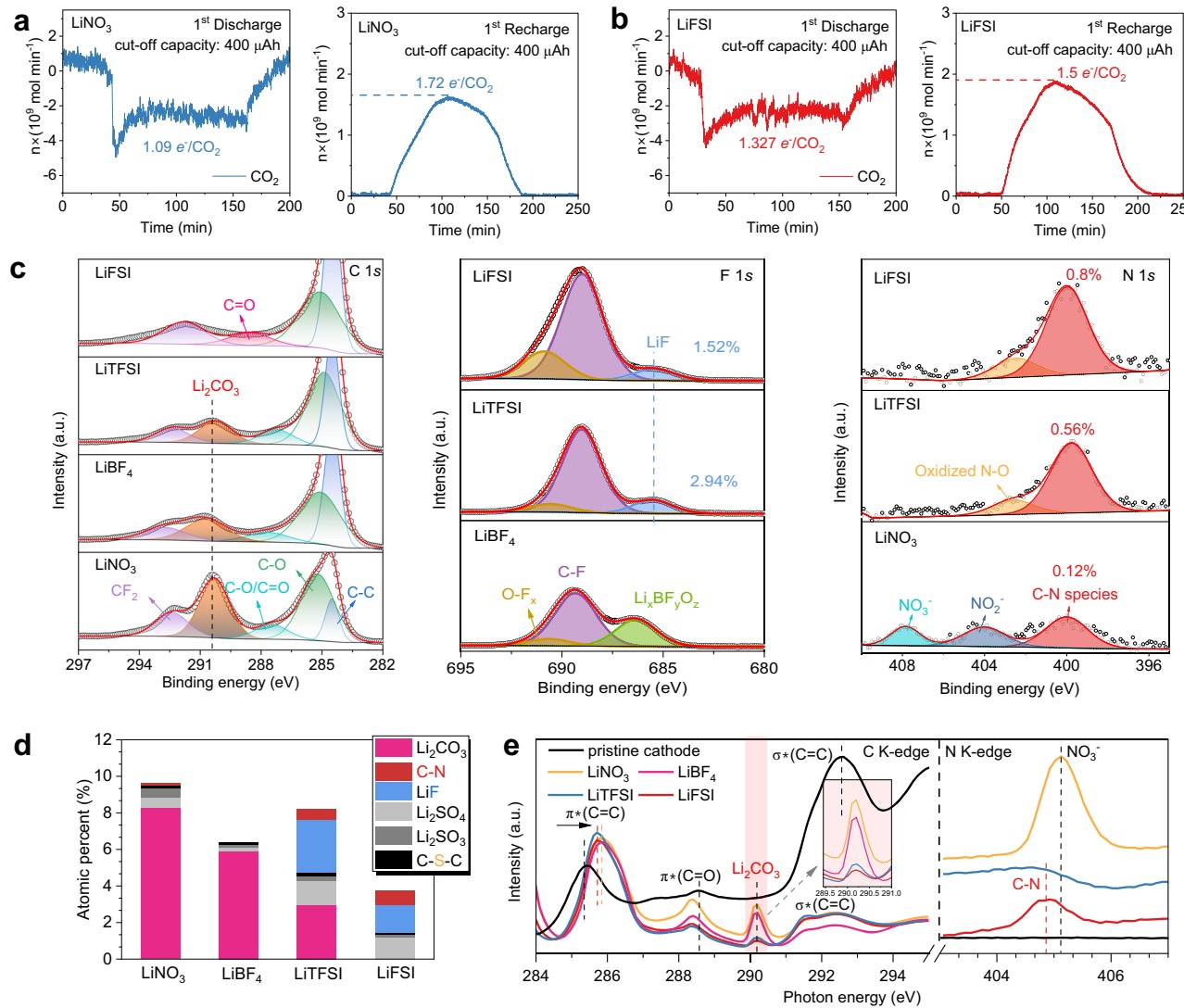

**Fig. 3 | Characterization of discharge products and SEI composition on the cathodes.** In-situ DEMS analyses of Li-CO₂ cells during first discharge/charge in **a** LiNO₃ and **b** LiFSI electrolytes, tested at 200 μA within a capacity of 400 μAh. **c** C 1s, F 1s, and N 1s XPS spectra for the cathodes after 3 cycles (Li₂CO₃: 290.6 eV, LiF: 685.5 eV, NO₂⁻: 403.8 eV, and NO₃⁻: 408.1 eV, respectively; the bonding

configuration of pyrrolic-N at ~400.1 eV is in accord with C-N species)[54]. **d** The compounds assigned to the C, N, F, and S elements and their relative amounts. **e** C K-edge and N K-edge XANES spectra for the pristine cathode and cathodes after 3 cycles. The peak of Li₂CO₃ was highlighted by red shade with enlargement in the range of 289.5–291 cm⁻¹ in the inset.

confirmed the composition of SEI, mainly consisting of LiF and C-N species (Fig. 3d). Indeed, it is generally known that LiF as a critical SEI component is beneficial to electrode stability and battery cycling[15–19]. With lower LiF content (1.52%) on the cycled cathodes, however, it was surprisingly found that the cells in LiFSI electrolytes show better cycling stability than that in LiTFSI electrolytes with a higher LiF content (2.94%) on the cycled cathodes (Fig. 3c middle and Supplementary Fig. 25)[31]. This might offer a clue that the relative contents of C-N species in SEIs would affect the electrochemical performance to some extent, which is in good agreement with the calculated results related to the C-N species. As shown in Fig. 3c (right), the highest content of C-N species (0.8%) was detected on cathodes cycled in LiFSI cells, which delivers the longest cycle life. In comparison, a lower ratio of C-N species was found, 0.56% and 0.12% on cathodes cycled in LiTFSI and LiNO₃ cells, respectively, with both cells exhibiting relatively poor cycling performance[32–34]. Energy dispersive spectroscopy (EDS) results also demonstrate the same trend relating to the N content on the surfaces of cathodes (Supplementary Figs. 26–29), with the most prominent N signal in the LiFSI cell. These results indicate the catalytic

effects of C-N species towards promoting the decomposition of Li₂CO₃ (Fig. 3c) and thus prolonging cycle life (Fig. 2a). In addition, the relatively lower peaks of C-S-C suggest that the decomposition of electrolyte is effectively suppressed in LiFSI electrolytes (Supplementary Fig. 30)[16], consistent with the in-situ DEMS result (Fig. 3b).

To more deeply understand the effects of C-N species in SEIs on the cycling stability of Li-CO₂ batteries, X-ray absorption near-edge spectroscopy (XANES) was employed to trace the residuals of Li₂CO₃ and C-N species on cathodes after 3 cycles due to its high sensitivity for carbon detection[35]. As shown in Fig. 3e, the C K-edge XANES spectra for the LiNO₃ and LiBF₄ cells displayed the strong peak of Li₂CO₃ at 290.2 eV on cathodes after cycling[36]. In addition, the graphite π* (C = C) transition on all cycled cathodes was observed with a slight shift to a higher photon energy compared to the pristine one, indicating an interaction between Li₂CO₃ residuals and rGO catalysts at surfaces. With the existence of C-N species, the cathode cycled in LiFSI-based electrolytes (red line) shows less shift of π* (C = C) peak and much reduced intensity of π*(C = O) peak than that in LiNO₃ cells (yellow line, without C-N species), indicating the weaker interaction between

$Li_2CO_3$ residuals and rGO catalysts, and possible strong interaction between $Li_2CO_3$ and C-N species[37], in good agreement with our calculation results. It should be noted that the intensities of $Li_2CO_3$ residuals gradually decreased with increased peak intensity of C-N species at 404.8 eV in the N K-edge XANES spectra[38], further confirming the catalytic effects of C-N species towards $Li_2CO_3$ decomposition. In particular, the cells in $LiNO_3$ and $LiBF_4$ electrolytes with low contents of C-N species show significantly high intensities of $Li_2CO_3$ residuals peaks (Fig. 3e, yellow and purple lines, respectively), suggesting their inferior reversibility of $Li_2CO_3$ and poor cycling stability. In contrast, the cells in LiFSI electrolytes with a relatively higher content of C-N species show the lowest intensities of $Li_2CO_3$ residuals (Fig. 3e, red lines), thus superior cycling performance. The XANES results are in accordance with the positive correlation between C-N species and $Li_2CO_3$ residuals in the XPS analysis (Fig. 3c). These observations identified the positive correlation between the content of C-N species in SEIs and reversibility of $Li_2CO_3$, revealing that C-N species may play a critical role in enhancing reversibility of $Li_2CO_3$ and battery cyclability, which is in good agreement with our calculated results.

### Relationship between electrolyte solvation structures and chemical composition of the SEI

Theoretical studies and experimental observations of electrolyte structures were employed to understand the origin of C-N species in SEIs in various electrolytes. Raman measurements were first conducted to study their solvation structures, showing various solvation structures in these electrolytes (Supplementary Fig. 31)[39], and providing the detailed information about the coordination environments of different anions. It has been widely accepted that a shift of an anion peak to higher wavenumbers in electrolytes indicates that the anion binds with $Li^+$ ions, and more shift suggests stronger $Li^+$-anion interaction[40]. With increasing shift, the interaction can be classified into free anion, solvent-separated ion pairs (SSIP), and contact ion pairs (CIP)[41]. As shown in Fig. 4a, $LiNO_3$ and LiFSI electrolytes show more CIP (52.5% and 44.65 %, respectively) relative to the other electrolytes, while a smaller proportion of CIP is observed in the others: LiTFSI (13.7 %) and $LiBF_4$ (12.4 %) (Supplementary Tables 2 and 3). This demonstrates that $NO_3^-$ and $FSI^-$ have the strongest coordination with $Li^+$ ions compared to the others, which can promote anion-derived SEI generation on electrodes.

Molecular dynamics (MD) and radial density function (RDF) investigations further reveal the structure of solvation shells and overall electrolyte configurations. In RDFs (Supplementary Fig. 32), the sharp peaks at around 2 Å for Li-anion coordination indicate the existence of all the anions in the inner solvation shell, and their intensities decrease in the order of $LiNO_3$ > $LiBF_4$ > LiFSI > LiTFSI. The corresponding coordination numbers of anions confirm the trend, as shown in Fig. 4b (yellow columns). Taking $LiNO_3$ and LiFSI electrolytes as examples, the average coordination number is 2.31 for $LiNO_3$ versus 1.42 for LiFSI, indicating large amounts of anions in inner solvation shells. Furthermore, more information was provided by the surrounding environment of $Li^+$ solvation clusters, which can be classified as solvent-surrounded $Li^+$, $Li^+$-single anion pair, and $Li^+$-multiple anion cluster by coordination numbers of anions with $Li^+$ ions of 0, 1, and ≥2 (Supplementary Table 4). The most probable $Li^+$ solvation clusters are $Li^+$-single anion pairs and $Li^+$-multiple anion clusters for $LiNO_3$, $LiBF_4$, and LiFSI electrolytes (Fig. 4c–e), whereas solvent-surrounded $Li^+$ clusters dominate for the LiTFSI electrolyte (Fig. 4f). In particular, a large preference for $Li^+$ solvation clusters including anions (50% $Li^+$-single anion pairs and 36% $Li^+$-multiple anion clusters) is for LiFSI (Fig. 4f), which agrees with the high content of CIP (44.65 %) in the Raman results (Fig. 4a) and distinct $^{19}F$ NMR peaks from $FSI^-$ corresponding to the $Li$-$FSI^-$ coordination (Supplementary Fig. 33). Overall, both theoretical calculation and experiment results demonstrate that the solvation shells in $LiNO_3$ and LiFSI electrolytes contain more anions around $Li^+$ ions, which is promising for a preferentially anion-derived SEI on electrodes.

To explain why the content of C-N species in LiFSI is higher than that in $LiNO_3$ electrolytes (as demonstrated in Fig. 3c, right), density functional theory (DFT) calculations were performed to investigate the bonding strength between $Li^+$ ion and different anions. The binding free energy ($\Delta G_{bind}$) was calculated between one $Li^+$ ion and one anion to examine the bonding strength of different anions with an Li ion (Supplementary Fig. 34)[40]. All $Li^+$-anion complexes show negative $\Delta G_{bind}$ (Supplementary Fig. 35), and a more negative $\Delta G_{bind}$ suggests a stronger coordination of anions with $Li^+$ ion in the solvation shell. Compared to all the other anions, $NO_3^-$ shows the most negative $\Delta G_{bind}$ of −0.83 eV, whereas $FSI^-$ shows the least negative $\Delta G_{bind}$ of −0.43 eV (Fig. 4b, green columns). The relatively high $\Delta G_{bind}$ suggests that $FSI^-$ preferentially desolvates with $Li^+$ ion than other anions, and then decomposes on the cathode surface, leading to a high content of C-N species in SEI layer[13,42]. This reveals that the decomposition reactions not only depend on the solvation structure, but also the coordination capability between different components in inner solvation shells.

Based on the discussion in the characterization part, we believe that in-situ formed C-N species in SEIs could enhance the reversibility of $Li_2CO_3$, thus improving the cycling performance of $Li$-$CO_2$ batteries. To achieve high content of C-N species in SEIs, electrolyte design for $Li$-$CO_2$ batteries may obey the following principles: First, the employment of electrolyte salt with nitrogen-containing anions is the key to forming C-N species in SEI layers. Second, involvement of N-containing anions in $Li^+$ solvation shells, reflected by coordination numbers and surrounding environment of $Li^+$ solvation clusters, ensures the formation of anion-derived SEI layers containing C-N species. Finally, a high binding free energy between Li ion and anion, gives an estimate of coordination capability and weak Li-anion interaction in inner solvation shells, suggesting preferentially anion-derived SEI formation.

### Synergistic effect of C-N species and LiF components of SEI in dual-salt electrolytes

Given the electrolyte design principles discussed above, we developed 1 M dual-salt (including $LiNO_3$ and LiTFSI, $LiBF_4$ and LiTFSI, $LiNO_3$ and LiFSI, $LiBF_4$ and LiFSI, as shown in Supplementary Fig. 36) electrolytes in expectation of further improvements in battery performance through increasing the involvement of N-containing anions in solvation shells to generate a C-N rich SEI on cathodes. We found that the cells using $LiNO_3$/LiFSI dual-salt electrolytes perform a distinct improvement in electrochemical performance. Notably, the cell in 0.25 M $LiNO_3$/0.75 M LiFSI electrolytes exhibits the highest amount of C-N species (Fig. 5c, d) and delivers the best cycling performance, which is in good agreement with our rationale.

With the addition of $LiNO_3$, there is substantial improvement in battery performance (Supplementary Fig. 37). The $Li$-$CO_2$ cell in an optimized dual-salt electrolyte with 0.25 M $LiNO_3$/0.75 M LiFSI can be charged/discharged up to 150 cycles (1500 h) (Fig. 5a), significantly improving the cycle life compared to that using the best single-salt electrolyte (LiFSI, 900 h). Supplementary Fig. 38 displays the obvious superiority of dual-salt electrolytes in terms of full-discharge capacity, especially the highest capacity of 15,127.9 mA h $g^{-1}$, which was realized in the 0.25 M $LiNO_3$/0.75 M LiFSI cell. The superior performance of dual-salt electrolytes was also demonstrated by their rate capability (Supplementary Fig. 39), in which the 0.25 M $LiNO_3$/0.75 M LiFSI cell exhibits a much lower overpotential than the single-salt LiFSI cell among all the current densities (Supplementary Fig. 40). Moreover, electrode with Ru nanoparticles without further modification as catalysts was applied to lower the overpotential and enhance the cyclability of $Li$-$CO_2$ batteries, it performed 2200 h of cycling with 1 V overpotential in the first 40 cycles (Supplementary Fig. 41). Overall, batteries with the 0.25 M $LiNO_3$/0.75 M LiFSI electrolyte show improved battery lifespan, reduced overpotential, and lower

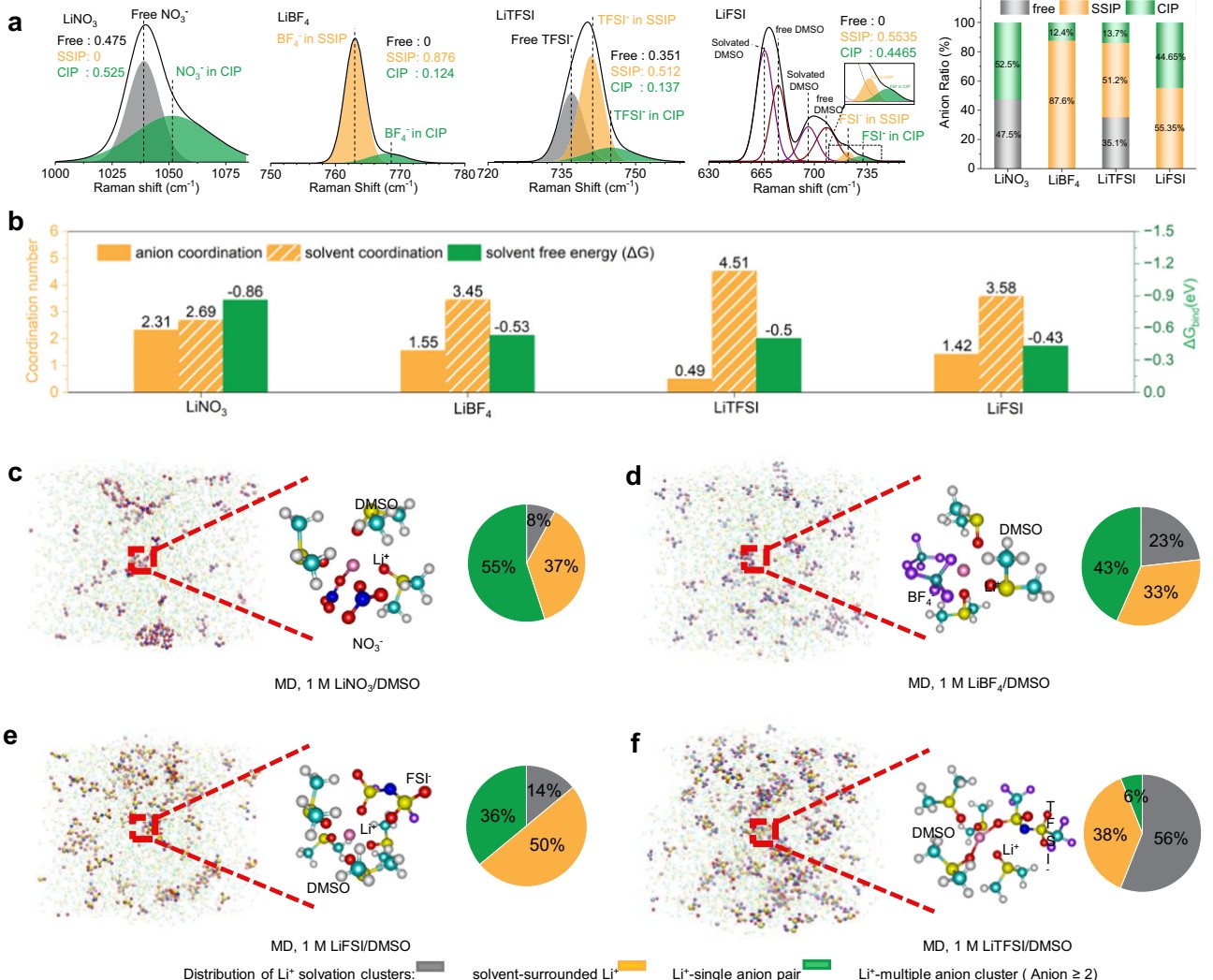

**Fig. 4 | Theoretical and experimental studies on the Li$^+$ solvation structures and electrolyte configurations. a** Fitted Raman curves of anions for different anion pairs of free anions, SSIP (solvent-separated ion pairs), and CIP (contact ion pairs), and their ratio comparison, with enlargement in the range of 710–745 cm$^{-1}$ for the LiFSI electrolyte in the inset. The peaks located at 667 and 697 cm$^{-1}$ can be assigned to the C-S symmetric and asymmetric stretching vibrations of free DMSO, respectively. The peaks at 676 and 708 cm$^{-1}$ are attributed to the C-S symmetric and asymmetric stretching vibrations of DMSO molecules that solvate with Li$^+$

ions[17,39,55,56]. **b** Left Y-axis: coordination numbers of different anions (yellow) and DMSO solvent (yellow slashes) in inner solvation shells by MD; right Y-axis: Binding free energy $\Delta G_{bind}$ between Li$^+$ and different anions calculated by DFT. MD simulation boxes, representative structures of Li$^+$ solvation clusters, and the electrolyte configurations based on the percentages of different Li$^+$ solvation clusters for **c** LiNO$_3$, **d** LiBF$_4$, **e** LiFSI, and **f** LiTFSI electrolytes. Colour scheme of molecules: Li, pink; C, light blue; H, white; O, red; N, navy; S, yellow; F, purple; and B, blue.

estimated cost due to relative low price of LiNO$_3$ (Supplementary Table 5), suggesting its suitability as electrolyte for Li-CO$_2$ batteries (Fig. 5b).

To verify the positive effect of C-N species towards enhanced battery performance in dual-salt electrolytes, the correlation between the reversibility of Li$_2$CO$_3$ and the content of C-N species in SEIs was first analysed. After 3 cycles, the XANES spectra show that the intensities of Li$_2$CO$_3$ residuals in the C K-edge spectra decreased with increased peak intensity of C-N species in the N K-edge spectra (Fig. 5c). More importantly, no Li$_2$CO$_3$ residuals in the C K-edge XANES spectrum could be detected in the 0.25 M LiNO$_3$/0.75 M LiFSI cell, which displays the most distinguish peak of C-N species and superior cycling performance. The positive effect of C-N species towards battery cycling can be also confirmed by XPS spectra, displaying no Li$_2$CO$_3$ residuals in cells using dual-salt electrolytes, even after 50 cycles (Supplementary Figs. 42 and 43). Compared to the XPS results of single-salt LiFSI electrolytes (0.8% C-N species), the XPS results of

dual-salt electrolytes after 3 cycles present a relatively higher content of C-N species (1.1% for the 0.25 M LiNO$_3$/0.75 M LiFSI cell and 0.89% for the 0.75 M LiNO$_3$/0.25 M LiFSI cell) (Supplementary Fig. 44), consistent with their relatively long cycle life. These results confirm the catalytic effect of C-N species towards promoting Li$_2$CO$_3$ decomposition and prolonging battery cycling. Interestingly, except for increased content of C-N species, we found that relatively higher amounts of LiF were observed on the cathodes cycled in the cells using dual-salt electrolytes (Supplementary Figs. 45 and 46) compared with that of the single-salt LiFSI cell (Fig. 3c, middle). This suggests less electrolyte consumption owing to the effective protection to electrodes from LiF-rich SEI layers, which can be evidenced by the lowest intensity of the by-products from solvent (DMSO) decomposition, such as C-S-C (Supplementary Fig. 47). In-situ DEMS results further proved better reversibility of Li$_2$CO$_3$ and less electrolyte decomposition in cells using dual-salt electrolytes. Taking the 0.25 M LiNO$_3$/0.75 M LiFSI electrolytes as an example, its $e^-$/CO$_2$ was determined to be 1.3 during

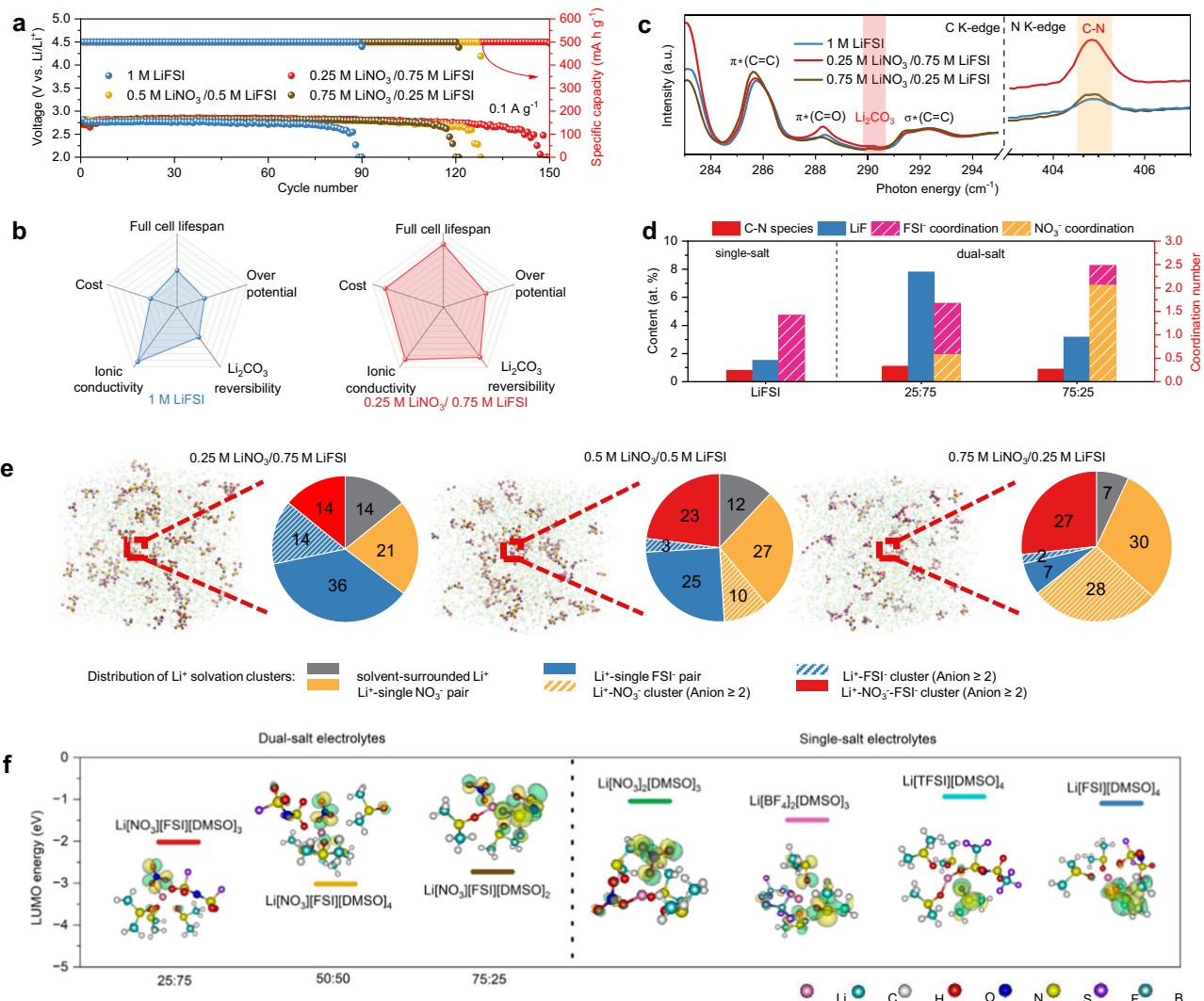

**Fig. 5 | Summary and overall evaluation of dual-salt electrolytes. a** Long-term cycling performance at 0.1 A g⁻¹ with a cut-off capacity of 500 mAh g⁻¹ (in the cut-off voltage from 2 V to 5 V). **b** Overall comparison of the properties and performances of the LiFSI electrolyte and 0.25 M LiNO₃/0.75 M LiFSI electrolyte. **c** C K-edge and N K-edge XANES spectra for the cathodes after 3 cycles in single-salt LiFSI and dual-salt electrolytes. **d** Comparison of C-N species and LiF content on cathodes after 3 cycles in single-salt LiFSI and dual-salt electrolytes, and their coordination numbers of anions for these electrolytes. **e** MD simulation boxes and the corresponding percentages of different Li⁺ solvation clusters in dual-salt electrolytes. **f** LUMO energy values for representative Li⁺ solvation clusters in single-salt and dual-salt electrolytes. Colour scheme of molecules: Li, pink; C, light blue; H, white; O, red; N, navy; S, yellow; F, purple; and B, blue. The yellow and blue zones represent electron loss and gain, respectively.

recharging (Supplementary Fig. 48). It is very close to the theoretical value of 1.33 (4e⁻/3CO₂) as the reversible decomposition of Li₂CO₃, indicating its excellent reversibility of Li₂CO₃ and high electrolyte stability. As summarized in Fig. 5d and Supplementary Fig. 49, C-N and LiF-rich SEI layers on cathodes were constructed in dual-salt electrolytes, which can not only achieve a highly reversible Li₂CO₃ formation/decomposition behaviour but also effectively suppress electrolyte consumption, thus achieving improved cycling performance of Li-CO₂ battery via the synergistic effect of C-N species and LiF components in SEI layer. To better offer direct evidence of the positive effect of C-N species towards battery cycling, we have conducted additional work comparing the cycling performance of Li-CO₂ cells using uncycled cathodes and cathodes, after the formation of C-N species in SEI layers (the cathodes after 5 cycles in the Li-CO₂ cells using the optimized 0.25 M LiNO₃/0.75 M LiFSI electrolytes, denoted as cathodes with C-N species). This was done using conventional electrolytes (1 M LiTFSI/DMSO and 1 M LiTFSI/TEGDME, which have been widely used in Li-CO₂ batteries). To eliminate the possible effects from the passivation of the

Li anode by electrolytes, all the cells were paired with fresh Li metal. When paired with cathodes with C-N species, substantial improvement in battery cycling occurred, even in conventional electrolytes, as summarized in Supplementary Table 6, Supplementary Figs. 50 and 51.

To understand the origin of increased contents of C-N species and LiF in dual-salt electrolytes, MD simulations and DFT calculations were jointly employed to unveil the SEI formation mechanisms. MD simulations predict the existence of (Li⁺-NO₃⁻-FSI⁻) clusters in dual-salt electrolytes (Fig. 5e), which present a reduced lowest unoccupied molecular orbital (LUMO) energy (from −2.02 to −3.03 eV) than (Li⁺-anions) in single-salt electrolytes (from −0.93 to −1.49 eV) (Fig. 5f and Supplementary Fig. 52). This indicates that the anions in these clusters exhibit an increased regional electrophilicity due to the participation of NO₃⁻ in solvation shells, leading to their higher possibility of accepting electrons[43]. Therefore, the decomposition of FSI⁻ in (Li⁺-NO₃⁻-FSI⁻) solvation clusters can be greatly promoted, explaining the increased amounts of C-N species and LiF in dual-salt electrolytes (Fig. 5d). In addition, we found that coordination numbers of anions

($NO_3^-$ and $FSI^-$) in dual-salt electrolytes are higher than that for the single-salt LiFSI electrolytes (Supplementary Fig. 53), suggesting their greatly increased nitrogen-containing anions in solvation shells. More importantly, compared to the other dual-salt electrolytes, 0.25 M $LiNO_3$/0.75 M LiFSI electrolytes show the largest coordination number of $FSI^-$ anions (1.07) (Fig. 5d, purple slashes columns) and the highest proportion of the clusters including $FSI^-$ (36% $Li^+$-single $FSI^-$ pairs + 14% $Li^+$-multiple $FSI^-$ clusters + 14% $Li^+$-$NO_3^-$-$FSI^-$ clusters) (Fig. 5e, blue and red pies, and Supplementary Table 7). This suggests its highest amounts of C-N species and LiF in SEIs, consist with our experimental XPS results (Fig. 5d, red and blue columns). Benefited from the anion-derived dense SEI layers formed on cathodes, SEI layer on cathodes in dual-salt electrolyte, exhibits enhanced catalytic and protective properties, therefore, $Li$-$CO_2$ battery exhibits improved overall electrochemical stability compared to that in single-salt electrolytes.

## Discussion

In summary, the critical roles of organic C-N species in facilitating the $CO_2$ reaction kinetics and achieving high reversibility of $Li_2CO_3$ has been demonstrated in this work. Through manipulating the SEI components on cathodes, the experimental results revealed the positive correlation between C-N species in the SEI and battery cycling, as demonstrated by the fact: an LiFSI-based cell with relatively high C-N species and low LiF in its SEI layer could deliver superior cycling performance to that of a LiTFSI-based cell with low C-N species and high LiF. Further theoretical calculations revealed the underlying mechanism: C-N species show strong adsorption towards $CO_2$ and $*CO_2^{2-}$ to promote initial activation of the reaction and charge transfer in the CRR process, driving spontaneous and fast reduction reaction kinetics. In addition, the strong interaction between C-N species and $Li_2CO_3$ could build a bridge to enable fast charge transfer and high catalytic activity in the CER, which can further effectively enhance the $Li_2CO_3$ decomposition kinetics and thus the cycling performance. Dual-salt ($LiNO_3$/LiFSI) electrolytes were designed to further improve cyclability through increasing the contents of C-N species. The 0.25 M $LiNO_3$/0.75 M LiFSI cell exhibits stable cycling over 220 cycles (2200 h) with 1 V overpotential. Our findings provide a deep understanding of the correlation between the organic SEI components and battery performance, thus offering an electrolyte design principle to overcome the critical issues facing $Li$-$CO_2$ batteries for future applications.

## Methods
### Preparation of electrolytes
Dimethyl sulfoxide (DMSO) (Sigma-Aldrich, 99.9%) was dried twice over freshly activated 4 Å molecular sieves. Lithium nitrate ($LiNO_3$, ≥ 99.0%, ReagentPlus®, Sigma-Aldrich), lithium tetrafluoroborate ($LiBF_4$, 98%, Sigma-Aldrich), lithium bis(trifluoromethanesulfonyl) imide (LiTFSI, 99.95%, Sigma-Aldrich), and lithium bis(fluorosulfonyl) imide (LiFSI, 99.5%, Canrd) were dried under vacuum at 80 °C overnight. All electrolytes were prepared and stored in an Ar-filled glovebox ($O_2$ and $H_2O$ levels < 0.01 ppm). All the electrolytes were composed of 1 M lithium salt(s) in DMSO solvent.

### Preparation of cathodes
The air cathodes were prepared via a filtration process. Typically, 5 mg of catalyst materials (reduced graphene oxide, rGO) and 50 μL of Nafion solution (~5 wt%) were dispersed in 2 mL of ethanol. After being ultrasonicated for around 60 min, the suspension was filtered using Toray carbon paper (TGP-H-060) as the filtering paper. After being dried at 80 °C overnight, the catalyst was uniformly coated on the Toray carbon paper. The catalyst-loaded Toray carbon paper was then punched out into circular sheets with a diameter of 9 mm, which were directly used as air cathodes. Here, the Toray carbon paper served as the gas diffusion layer and current collector for the air cathodes.

### Electrochemical measurements
For electrochemical tests, CR2032-type coin cells (16 holes on the cathode side) were assembled in an Ar-filled glove box with air electrodes and lithium chip anodes separated by a glass fibre separator (Whatman, diameter: 19 mm). Solutions of 1 M lithium salts in 1 L DMSO solvent were used as electrolytes. The as-prepared coin cells were sealed in $CO_2$-filled bottles for $Li$-$CO_2$ battery tests. The galvanostatic discharge/charge tests were carried out with high cut-off voltage of 5 V and low cut-off voltage of 2 V at various current densities (from 0.1 to 2 A g$^{-1}$) using a battery test station (Land, China) at 25 °C. Electrochemical impedance spectroscopy (EIS) was performed over the frequency range of 100 kHz to 10 MHz with a perturbation amplitude of ±10 mV using a VMP3 potentiostat/galvanostat (BioLogic).

### Characterization
Raman spectra were collected on a Raman spectrometer (HORIBA LabRAM HR Evolution) using the 532 nm line of a semiconductor laser (100 MW, frequency-doubled Nd: YAG laser from Laser Quantum). The attenuated total reflection (ATR)-FTIR spectra were obtained using a Nicolet 6700 ThermoFisher instrument. The morphologies of the cathodes were investigated using field-emission scanning electron microscopy (FESEM, FEI QUANTA 450 FEG) and a cryo-transmission electron microscope (TEM). The cryo-TEM characterization was performed on a Thermo Scientific Glacios microscope with an accelerating voltage of 200 kV. High-resolution TEM images were obtained using a Thermo Scientific Falcon 4 camera at a dose rate of ~6 e$^{-1}$ px$^{-1}$ s$^{-1}$ with a dosage of ~40 eÅ$^{-2}$. X-ray photoelectron spectroscopy was conducted to analyse the chemical composition on the surface of the pristine and cycled cathodes on a VG Multilab 2000 (VG) photoelectron spectrometer using monochromatic Al Kα radiation under vacuum of $2 \times 10^{-6}$ Pa. Ex-situ synchrotron X-ray absorption spectroscopy was carried out at the soft X-ray beamline, Australian Synchrotron. The conductivity of electrolytes was obtained from a conductivity measurement instrument (DDB-303A, Shanghai INESA & Scientific Instrument Co. LTD).

### Theoretical calculations
All MD simulations were performed in the GROMACS package using the Optimized Potentials for Liquid Simulations (OPLS) force field. ACPYPE[44] was employed to obtain the force field topology along with the previously developed force field topology for $BF_4^{-}$[45]. A simulation box size of $5 \times 5 \times 5$ nm$^3$ was used in all simulation models. Further details on the ratios of electrolyte components and MD simulation parameters are available in the Supporting Information.

DFT calculations were implemented using the Vienna ab-initio simulation package (VASP 5.4.4)[46,47] with the core and valence electronic interactions being modelled using the projector augmented wave (PAW) method[48,49]. The Perdew-Burke-Ernzerhof (PBE) exchange-correlation function was employed[50]. Molecular orbital energy levels (*i.e.*, LUMO) and different species interactions in solution were calculated using VASP with a kinetic energy cut-off of 500 eV and a single Gamma k-point. For the interaction between $Li^+$ and other components in the electrolyte, including anion and solvent molecules, the thermodynamic cycle presented in Supplementary Figs. 34 and 35 was used, with the following Eq. 1:

$$\Delta G_{bind} = -G_{DFT} + G_{Livap} + G_{Liion} + G_{Lisolv} \qquad (1)$$

where $G_{DFT}$, $G_{Li\,vap}$, $G_{Li\,ion}$, $G_{Li\,solv}$ represent the electronic energy obtained from DFT calculations, Li vaporization energy from solid to gas, Li ionization energy from the atomic to the ionic state, and the approximation of Li ion solvation energy in the organic solvent[51,52].

For the interaction between different species and graphene as an electrode, a vacuum region of 20 Å was introduced in the direction of the z-axis to avoid interactions between periodic images.

The wavefunction was expanded with a kinetic energy cut-off of 500 eV, and a Gamma k-point mesh of $3 \times 3 \times 1$ was used. Geometrical optimizations were achieved by relaxing all ionic positions and supercell vectors until the Hellmann-Feynman forces were less than 0.01 eV Å$^{-1}$. The dispersion correction was also considered in this study by using the DFT-D3 method[53]. The adsorption energy was calculated by using the following Eq. 2:

$$E_{ads} = E(\text{graphene} * \text{adsorbate}) - E(\text{graphene}) - E(\text{adsorbate}) \quad (2)$$

where graphene represents all different functional graphene investigated here (Fig. S2) and adsorbates include Li, $CO_2^{2-}$, $CO_3^{2-}$, and $Li_2CO_3$. The zero-point energy and entropic contributions at 298 K as well as the solvation correction for $CO_2$ were considered in the calculations of adsorption energy.

## Reporting summary

Further information on research design is available in the Nature Portfolio Reporting Summary linked to this article.

## Data availability

Source data are provided with this paper. All other data are available from the authors upon reasonable request. Source data are provided with this paper.

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

## Acknowledgements

Financial support from the National Natural Science Foundation of China (Grant No. 52104315, W.Z.), the National Key R&D Program of China (No. 2022YFC3900200, W.Z.), and the Australian Research Council (DP210101486 and FL210100050, Z.G.) is acknowledged. Part of this work was carried out at the Soft X-ray (SXR) beamline (beamtime: M18876, G.L.). The authors acknowledge their operational support from ANSTO staff for synchrotron-based characterizations.

## Author contributions

F.Z. performed the experiments and wrote the original draft. J.A.Y. performed the theoretical calculations. J.Z. and L.S. performed the catalyst preparation and testing. Y.F. and G.L. conducted part of the cathode characterizations and analysed the data. W.Z. and Z.G. directed the project and revised the manuscript. D.W. provided helpful discussions and English editing.

## Competing interests

The authors declare no competing interests.
