## [Peer Review File · Nature Communications]

Catalytic Role of in-situ formed C-N Species for Enhanced Li₂CO₃ DecompositionREVIEWER COMMENTS

Reviewer #1 (Remarks to the Author):

In this manuscript, the authors report the catalytic effect of C-N species from electrolyte species on the decomposition of Li_2CO_3 in Li-CO₂ batteries. They carried out a rather detailed study of the battery reaction process, electrolyte solvation structure and potential catalytic mechanism. While the designed electrolyte and the interfacial species formed show improved battery performance, the claimed reaction mechanism is not reasonably substantiated. There are several critical questions that need to be addressed before giving the final recommendation about this work. Therefore, the reviewer suggests rejection and resubmission for this manuscript. The concerns are listed below.

1. First of all, it is unclear whether the improved cycling performance is due to the passivation of the Li anode by LiFSI, which has been widely used to stabilize Li metal. The authors are suggested to use a stable "counter" electrode, for example, LFP/LTO, or replace the cycled Li anodes to verify whether the cycle life difference is from the cathode.
2. To claim the catalytic effect of the C-N species on Li_2CO_3 decomposition, the C-N species need to be relatively persistent instead of reforming during each cycle. How would the battery perform if the electrolyte was replaced by a conventional electrolyte after the formation of C-N species during the initial cycle?
3. The authors used DEMS to measure the amount of CO₂ evolution during charging. The reviewer would suggest quantifying the amount of CO₂ consumption during discharge as well. It is also well-known that LiFSI is very reactive with nucleophilic species. Would the discharge/charge intermediates continuously consume LiFSI? Or to what extent? DMSO is also regarded as unstable under high voltages. The side reactions related to DMSO also need to be carefully examined.
4. The existence and the identity of the C-N species is not well supported. The binding energy of N 1s would be significantly influenced by the chemical environment and the oxidation state of N atoms. With the high cut-off voltage at 4.5 V (not specified in the experimental part), it is unclear whether the N atoms would be bonded with O=S=O moieties (as its origin in the LiFSI salt) or in other chemical environments. In this regard, the C-N model would be over-simplified to deviate from the actual composition.
5. It is very strange that the bonding between Li⁺ ions and anions calculated by the authors shows positive free energy changes. Such electrostatic attractions are naturally favorable as found in their salt forms.
6. The authors claim the dominance of CIP solvation species in the LiFSI-electrolyte. However, it is generally believed that LiFSI is highly dissociating and it is unlikely to favor CIP in a highly polar solvent like DMSO.

Reviewer #2 (Remarks to the Author):

This paper uncovers the significant contribution of C-N species within the SEI layer. These species play a crucial role by offering adsorption sites and facilitating rapid charge transfer to *CO₂⁻ and Li_2CO_3 , thereby effectively controlling the reversibility of Li_2CO_3 throughout battery cycling. Furthermore, the authors have successfully engineered a C-N enriched SEI layer within a dual salts electrolyte, resulting in sustained long-term cycling performance. This achievement offers fresh perspectives on SEI structure, electrolyte formulations, and the design of catalytic active sites in Li-CO₂ batteries. However, it is still not clear to me the dynamics of the key redox reaction of $3\text{CO}_2 + 4\text{Li}^+ + 3\text{e}^- \leftrightarrow 2\text{Li}_2\text{CO}_3 + \text{C}$ and effects of

lithium salt LiFSI, more evidence and explanation would be recommended. Overall, I would like to reconsider its publication if the following concerns can be addressed.

1. As the $\cdot\text{CO}_2^-$ is a key intermediate radical from CO_2 to Li_2CO_3 and authors do some FTIR work to demonstrate the formation of Li_2CO_3 in Fig. S11, I wonder if it's more straightforward to confirm the existence of $\cdot\text{CO}_2^-$ during reactions by powerful FTIR technique.
2. Given that the authors have employed multiscale characterizations, including FTIR, Raman, TEM, and XPS, to substantiate the presence of Li_2CO_3 , it raises the question of whether the stability of the cathode materials was considered during exposure to air or during the sample transfer process after cycles.
3. Based on above two comments, I wonder if in situ FTIR or Raman experiments can be done to prove the dynamics, for example, the existence of $\cdot\text{CO}_2^-$ and Li_2CO_3 , during cycling.
4. How to understand the difference in EIS for all four salts used in the paper? It appears that the EIS results for LiNO_3 , LiTFSI, and LiBF_4 do not exhibit significant distinctions. As the distinct EIS data from LiFSI is used to explain the superior performance, it's recommended to clarify how the variations in the performance of other salts are explained despite the apparent similarities in their EIS data?
5. Please enhance the image quality in Fig. S10, as the current quality does not convincingly confirm the presence of Li_2CO_3 . The existing image lacks the clarity needed to discern any lattice structure, and it is challenging to identify the specific features that the authors intend to emphasize in the TEM images. Additionally, the yellow circles drawn by the authors seem to obscure valuable information from the FFT, and I am uncertain whether I can concur with the authors' conclusions regarding this matter.
6. What's the primary distinction between lithium salts LiFSI and LiTFSI? It's worth noting that LiTFSI can also offer enhanced anion coordination and contains a similar nitrogen element to LiFSI. This raises questions about why a single LiFSI salt is more effective at facilitating the formation of C-N species within the SEI, and this aspect should be clarified for better understanding.
7. As LiTFSI can also provide more anion coordination and contains the similar N element with LiFSI. I wonder why single LiFSI salt can promote the formation of C-N in the SEI should be clearer.
8. What is the role or function of the Ru nanoparticles? The introduction of Ru nanoparticles in the paper appears somewhat abrupt. It would be beneficial to clarify the purpose of these nanoparticles and provide information on the electrode's performance in the absence of Ru nanoparticles.
9. Will the SEI continue to undergo cycles of collapse and reconstruction after multiple charge-discharge cycles? How can the effective function of C-N species be ensured throughout these cycles?
10. As the pristine cathode contains a wide variety of C-SO₃ species according to XPS in Fig S13, how can authors distinguish Li_2CO_3 and C-SO₃ as they share the similar binding energy?

Reviewer #3 (Remarks to the Author):

This manuscript by Guo et al. investigated the effect of C-N species in the formed solid electrolyte interphase (SEI) on the generation and decomposition of Li_2CO_3 during discharge and charge processes of Li-CO₂ battery. Theoretical analysis and experimental tests showed that C-N species enable strong adsorption sites and promote charge transfer

from interface to CO_2^- during discharge, and from Li_2CO_3 to C-N species during charge. The results can provide useful information on tuning the component of SEI for improved cycling stability of Li-CO₂ batteries. Before possible consideration of publication in Nat. Commun., the authors need to address the following issues.

- 1) Why did the authors choose to use rGO with poor catalytic performance as a matrix to investigate the role of the SEI components?
- 2) There are some format issues that can be corrected. For example, in the caption of X axis in Fig. 2e, "Cycles" can be changed as "Cycle"; the first letter of the words except the first word can be lower-case; among others.
- 3) Why did the overpotential jump at 2 A g⁻¹ especially for LiTFSI as shown in Fig. 2e?
- 4) For the sake of comparison, some spectroscopy measurements may also be conducted prior to battery cycle.
- 5) In Fig. 4a, the width of the yellow peak seems to be too large.
- 6) Are there some experimental proofs for the formation of Li-N bonds?
- 7) What's the effect of different C-N configurations on the catalytic performance?

Point-by-Point Response to the Reviewers' Comments

Reply to Reviewer #1

Overall comment: In this manuscript, the authors report the catalytic effect of C-N species from electrolyte species on the decomposition of Li_2CO_3 in Li-CO₂ batteries. They carried out a rather detailed study of the battery reaction process, electrolyte solvation structure and potential catalytic mechanism. While the designed electrolyte and the interfacial species formed show improved battery performance, the claimed reaction mechanism is not reasonably substantiated. There are several critical questions that need to be addressed before giving the final recommendation about this work. Therefore, the reviewer suggests rejection and resubmission for this manuscript. The concerns are listed below.

Reply: We value your comments and suggestions, which have been very helpful for improving the quality of our manuscript, particularly via the request for improved justification of the proposed reaction mechanism. To achieve this, we have conducted more battery tests, and obtained more experimental evidence which is also consistent with the proposed reaction mechanism. We have also corrected the mistakes in the revised manuscript, according to your kind comments. We sincerely hope that the manuscript will meet with your approval after revision.

Comment 1: First of all, it is unclear whether the improved cycling performance is due to the passivation of the Li anode by LiFSI, which has been widely used to stabilize Li metal. The authors are suggested to use a stable "counter" electrode, for example, LFP/LTO, or replace the cycled Li anodes to verify whether the cycle life difference is from the cathode.

Reply: Thank you for your valuable suggestion. Li-CO₂ batteries as a typical metal-CO₂ battery are constituted by a lithium anode, electrolyte, ion-conducting separator, and a porous cathode that can breathe CO₂. [1-4] Battery operation involves the absorption of CO₂ that reacts with Li⁺ ions migrating from the Li anode to form discharge products during discharge and, during charging, their reversible decomposition to release CO₂. Operation is based on the reaction: $3\text{CO}_2 + 4\text{Li}^+ + 4\text{e}^- \leftrightarrow 2\text{Li}_2\text{CO}_3 + \text{C}$. [5,6], and because of this the Li anode cannot simply be replaced by other anode materials, such as LFP/LTO, in Li-CO₂ batteries.

To better offer direct evidence of the positive effect of C-N species towards battery cycling, we have conducted additional work comparing the cycling performance of Li-CO₂ cells using uncycled cathodes and cathodes, after the formation of C-N species in SEI layers (the cathodes after 5 cycles in the Li-CO₂ cells using the optimized 0.25 M LiNO₃/0.75 M LiFSI electrolytes, denoted as cathodes with C-N species). This was done using conventional electrolytes (1 M LiTFSI/DMSO and 1 M LiTFSI/TEGDME, which have been widely used in Li-CO₂ batteries). To eliminate the possible effects from the passivation of the Li anode by electrolytes, all the cells were paired with fresh Li metal. When paired with cathodes with C-N species, substantial improvement in battery cycling occurred, even in conventional electrolytes, as summarized in **Table S6**. In 1 M LiTFSI/DMSO electrolytes (**Fig. S50**), the cells using cathodes with C-N species performed with a much improved cyclability of 128 cycles with a slightly reduced overpotential of about 1.4 V at 0.1 A g⁻¹. In contrast, using uncycled cathodes only 82 cycles were achieved, with overpotential of 1.48 V. With further increases in current density to 0.3 A g⁻¹, the cells using cathodes with C-N species could also deliver prolonged cycle life (94 cycles) with a relatively low

overpotential of 1.76 V, compared to that using uncycled cathodes (52 cycles with overpotential of 2 V).

In 1 M LiTFSI/TEGDME electrolytes (Fig. S51), the cells using cathodes with C-N species exhibited an enhanced cycling performance of 68 cycles with a lower overpotential of about 2 V at 0.1 A g⁻¹, compared to that using uncycled cathodes (56 cycles and 2.15 V). When the current density was increased to 0.3 A g⁻¹, the cells using cathodes with C-N species could still run for 49 cycles, while that using uncycled cathodes suffered from a dramatic decay of discharge capacity. These electrochemical results confirm that the improved cycling performance of Li-CO₂ batteries in this work is mainly attributed to the cathode optimization, rather than Li anode stabilization.

Supplementary Table 6. Comparison of the cycling performance of Li-CO₂ cells using uncycled cathodes and cathodes with C-N species.

Conventional Electrolytes	Cathodes	Cycle life	Cycle life
		0.1 A g ⁻¹	0.3 A g ⁻¹
1 M LiTFSI/DMSO	Uncycled cathodes	80	82
	Cathodes with C-N species	128	94
1 M LiTFSI/TEGDME	Uncycled cathodes	56	-
	Cathodes with C-N species	68	49

Supplementary Fig. 50. Cycling performance of Li-CO₂ batteries in conventional 1 M LiTFSI/DMSO electrolytes (a) using uncycled cathodes, (b) using cathodes with C-N species, and (c) corresponding long-term voltage-time profiles at 0.1 A g⁻¹; (d) using uncycled cathodes, (e) using cathodes with C-N species, and (f) corresponding long-term voltage-time profiles at 0.3 A g⁻¹.

Supplementary Fig. 51. Cycling performance of Li-CO₂ batteries in conventional 1 M LiTFSI/TEGDME electrolytes (a) using uncycled cathodes, (b) using cathodes with C-N species, and (c) corresponding long-term voltage-time profiles at 0.1 A g⁻¹; (d) using uncycled cathodes, (e) using cathodes with C-N species, and (f) corresponding long-term voltage-time profiles at 0.3 A g⁻¹.

[1] Mu, X. et al. Li-CO₂ and Na-CO₂ batteries: toward greener and sustainable electrical energy storage. *Adv. Mater.* **32**, 1903790 (2020).

[2] Jiao, Y. et al. Recent progress and prospects of Li-CO₂ batteries: Mechanisms, catalysts and electrolytes. *Energy Storage Mater.* **34**, 148-170 (2021).

[3] Hu, A. et al. Design strategies toward catalytic materials and cathode structures for emerging Li-CO₂ batteries. *J. Mater. Chem. A* **7**, 21605-21633 (2019).

[4] Sun, X. et al. Recent advances in rechargeable Li-CO₂ batteries. *Energy & Fuels* **35**, 9165-9186 (2021).

[5] Sarkar, A. et al. Recent Advances in Rechargeable Metal-CO₂ Batteries with Nonaqueous Electrolytes. *Chem. Rev.* **123**, 9497-9564 (2023).

[6] Zou, J. et al. Revisiting the Role of Discharge Products in Li-CO₂ Batteries. *Adv. Mater.* 2210671 (2023).

Comment 2: To claim the catalytic effect of the C-N species on Li₂CO₃ decomposition, the C-N species need to be relatively persistent instead of reforming during each cycle. How would the battery perform if the electrolyte was replaced by a conventional electrolyte after the formation of C-N species during the initial cycle?

Reply: Thank you for this important question. To detect the variation of C-N species in SEI layers during cycling, we have provided XPS analysis results for the cathodes after 3, 10, 30, and 50 cycles in cells using single-salt LiFSI electrolytes. As shown in **Fig. S42a**, the N 1s XPS spectra indicate that there was almost no change of the content of C-N species with cycling, consistent with the persistent existence of C-N species with no evidence of reforming throughout the multiple discharge-charge cycles.

Supplementary Fig. 42. Characterization of SEI composition on the cathodes cycled in Li-CO₂ batteries. C 1s, N 1s, and S 2p XPS spectra, and the content comparison of C-N species and SO₄²⁻ for the cathodes after 3, 10, 30, and 50 cycles in the cells using (a) single-salt LiFSI electrolytes.

To further confirm the positive effect of C-N species formed on cathodes towards battery cycling, we have compared the cycling performance of Li-CO₂ cells using uncycled cathodes, and cathodes after the formation of C-N species in SEI layers (the cathodes after 5 cycles in the Li-CO₂ cells using the optimized 0.25 M LiNO₃/0.75 M LiFSI electrolytes, denoted as cathodes with C-N species) in conventional electrolytes (1 M LiTFSI/DMSO and 1 M LiTFSI/TEGDME, which have been widely used in Li-CO₂ batteries). To eliminate the possible effect from the passivation of the Li anode by electrolytes, all the cells were paired with fresh Li metal. When paired with cathodes with C-N species, a substantial improvement in battery cycling occurred, even if the electrolyte was replaced by a conventional electrolyte (Table S6). Detailed comparison has been provided above (Comment 1, Reviewer #1). This confirms that the C-N species in SEI layers on cathodes persistently play a catalytic role in enhancing battery cyclability, even in a conventional electrolyte.

Supplementary Table 6. Comparison of the cycling performance of Li-CO₂ cells using uncycled cathodes and cathodes with C-N species.

Conventional Electrolytes	Cathodes	Cycle life 0.1 A g ⁻¹	Cycle life 0.3 A g ⁻¹
1 M LiTFSI/DMSO	Uncycled cathodes	80	82
	Cathodes with C-N species	128	94
1 M LiTFSI/TEGDME	Uncycled cathodes	56	-
	Cathodes with C-N species	68	49

Comment 3: The authors used DEMS to measure the amount of CO₂ evolution during charging. The reviewer would suggest quantifying the amount of CO₂ consumption during discharge as well. It is also well-known that LiFSI is very reactive with nucleophilic species. Would the discharge/charge intermediates continuously consume LiFSI? Or to what extent? DMSO is also regarded as unstable under

high voltages. The side reactions related to DMSO also need to be carefully examined.

Reply: Thank you for your insightful suggestions. We have quantified the charge-to-mass ratios (e^-/CO_2) during discharge and provided relevant discussion in the revised manuscript.

The collected gases in the *in-situ* DEMS results show that CO_2 was consumed in first battery discharge, and CO_2 was released in first battery recharge (Figs. 3a, b). The theoretical value of e^-/CO_2 ratio is 1.33 ($4e^-/3\text{CO}_2$), corresponding to the decomposition of Li_2CO_3 in accordance with the redox reaction of $3\text{CO}_2 + 4\text{Li}^+ + 4e^- \leftrightarrow 2\text{Li}_2\text{CO}_3 + \text{C}$.²⁶ The deviation from the standard value of 1.33 indicates the occurrence of parasitic reactions. As shown in Fig. 3a, the e^-/CO_2 ratios in LiNO_3 cells deviated considerably from 1.33, which were determined as 1.09 and 1.72 for discharge and charge processes, respectively. In contrast, the ratios in the LiFSI cells were determined to be 1.327 and 1.5 during discharge and charge, respectively (Fig. 3b). The lower deviation from the standard e^-/CO_2 value in the LiFSI cell implies better reversibility of Li_2CO_3 and less electrolyte decomposition than that in the LiNO_3 cell, consistent with superior cycling performance of the LiFSI cell.

Fig. 3 *In-situ* DEMS analyses of Li- CO_2 cells during first discharge/charge in (a) LiNO_3 and (b) LiFSI electrolytes, tested at 200 μA within a capacity of 400 μAh .

We do agree that electron-deficient groups, such as S=O in the FSI⁻, are prone to degradation due to the nature of nucleophilic species (CO_2^{2-} in Li- CO_2 batteries, Equations S1-4).[1-2] This explains the formation of C-N species and LiF formed on cathodes in LiFSI electrolytes to some extent. The deviation of e^-/CO_2 (1.5) in LiFSI cells during first charge from the standard value of 1.33 also is consistent with decomposition of electrolyte (including LiFSI salt and DMSO solvent) in the initial cycle. However, it is well known that the protective SEI layers on electrodes resulting from FSI-reduction can effectively prevent continuous electrolyte consumption[3-5]. To detect the variation of C-N species with cycling, we compared the XPS analysis results for the cathodes after 3, 10, 30, and 50 cycles in cells using single-salt LiFSI electrolytes. As shown in Figs. S42a, the N 1s XPS spectra reveal almost no change in the content of C-N species with discharge/charge cycling. This suggests that LiFSI would not be continuously consumed by discharge/charge intermediates after the formation of SEI layers in the initial cycles, since LiFSI is the only source of nitrogen in electrolytes.

Equations S1-4:

Supplementary Fig. 42. Characterization of SEI composition on the cathodes cycled in Li-CO₂ batteries. C 1s, N 1s, and S 2p XPS spectra, and the content comparison of C-N species and SO₄²⁻ for the cathodes after 3, 10, 30, and 50 cycles in the cells using (a) single-salt LiFSI electrolytes.

By contrast, DMSO solvents as a nucleophilic oxidant are prone to react with nucleophilic species (CO₂²⁻) rather than LiFSI.[6-7] *In-situ* Raman spectra show that weak signals of DMSO₂ species (green dash line) were detected in LiFSI electrolytes (Fig. S20b), which suggests decomposition of DMSO solvents due to the attack of reduced *CO₂²⁻ radicals during the first cycling. It should be noted that the signals of DMSO₂ species gradually disappeared after full recharging. The S 2p XPS spectrum of cathodes after 3 cycles also displays the C-S-C peaks assigned to DMSO (Fig. S42a), indicating an incomplete decomposition of DMSO. With cycling, however, the S 2p XPS spectra (Fig. S42a) show that the content of Li₂SO₄ on cathodes suffers obvious variations. Considering almost no change in the content of C-N species (LiFSI not continuously consumed with cycling), the increase in the content of Li₂SO₄ possibly comes from the continuous decomposition of DMSO solvents.

Supplementary Fig. 20. In-situ Raman evidence for the Li-CO₂ cells with the formation/decomposition of Li₂CO₃ during discharge/charge. The *in-situ* Raman spectra collected at specific points (marked at voltage profile) during battery running at a constant current of 100 μ A with a cut-off capacity of 30 μ Ah for (a) LiNO₃ cells and (b) LiFSI cells. The peaks assigned to dimethyl sulfone (DMSO₂) species were observed during cycling. Corresponding voltage profiles are listed above each group of the *in-situ* Raman spectra.

[1] Garcia-Quintana, L. et al. Stabilisation of the superoxide anion in bis (fluorosulfonyl) imide (FSI) ionic liquid by small chain length phosphonium cations: Voltammetric, DFT modelling and spectroscopic perspectives. *Electrochem. commun.* **127**, 107029 (2021).

[2] Lai, J. et al. Electrolytes for rechargeable lithium–air batteries. *Angew. Chem. Int. Ed.* **59**, 2974–2997 (2020).

[3] Eshetu, G.G. et al. In-depth interfacial chemistry and reactivity focused investigation of lithium–imide-and lithium–imidazole-based electrolytes. *ACS Appl. Mater. Interfaces.* **8**, 16087-16100 (2016).

[4] Chen, J. et al. Electrolyte design for LiF-rich solid–electrolyte interfaces to enable high-performance micro-sized alloy anodes for batteries. *Nat. Energy* **5**, 386-397 (2020).

[5] Liu Y. et al. Self-assembled monolayers direct a LiF-rich interphase toward long-life lithium metal batteries. *Science* **375**, 739-745 (2022).

[6] Ottakam Thotiyil, M.M. et al. A stable cathode for the aprotic Li–O₂ battery. *Nat. Mater.* **12**, 1050-1056 (2013).

[7] Cao, D., Tan, C. and Chen, Y. Oxidative decomposition mechanisms of lithium carbonate on carbon substrates in lithium battery chemistries. *Nat. Commun.* **13**, 4908 (2022).

Comment 4: *The existence and the identity of the C-N species is not well supported. The binding energy of N 1s would be significantly influenced by the chemical environment and the oxidation state of N atoms. With the high cut-off voltage at 4.5 V (not specified in the experimental part), it is unclear whether the N atoms would be bonded with O=S=O moieties (as its origin in the LiFSI salt) or in other chemical environments. In this regard, the C-N model would be over-simplified to deviate from the actual composition.*

Reply: Thank you for your valuable comments. As shown in **Figure 1**, there are three possible atomic configurations of C-N bonds, corresponding to pyrrolic-N (denoted as C-N species, ~400.1 eV), pyridinic-N, and graphitic-N. In this work, however, pyrrolic-N (398.6 eV) and graphitic-N (401.4 eV) were not detected on the cycled cathodes based on the N 1s XPS spectra (**Fig.3c**). This eliminates the influence of oxidation state of N atoms on the identification of C-N species due to the absence of oxygen in its atomic configuration.

Figure 1. Top view of the atomic configurations of (a) pyrrolic-N, (b) pyridinic-N, and (c) graphitic-N. The carbon, oxygen, and nitrogen atoms are marked as copper, red, and purple, respectively. (d) Schematic illustration of nitrogen species with the corresponding reported XPS binding energies in the N-doped graphene.[1] (e) Adsorption energies of graphene, C-N (pyrrolic-N, pyridinic-N, and graphitic-N), C-F, C-O, and C-S species for CO₂, *CO₂²⁻, and Li₂CO₃.

Fig. 3 (c) C 1s, F 1s, and N 1s XPS spectra for the cathodes after 3 cycles (Li₂CO₃: 290.6 eV, LiF: 685.5 eV, NO₂⁻: 403.8 eV, and NO₃⁻: 408.1 eV, respectively; the bonding configuration of pyrrolic-N at ~400.1 eV is in accord with C-N species).

We are grateful for the reminder about the cut-off voltage. We have added the relevant information in the experimental section of the revised supplementary information. **The galvanostatic discharge/charge tests were carried out with high cut-off voltage of 4.5 V and low cut-off voltage of 2 V at various current densities (from 0.1 to 2 A g⁻¹) using a battery test station (Land, China) at room temperature.**

We have also provided XPS analysis for cathodes before and after cycling to trace the existence of C-N species and its variations with cycling. As shown in **Figs. S42a**, C-N species (~400.1 eV) were detected in the N 1s XPS spectra, without the S-N (162.1/163.3 eV) and O=S=O peaks (169.2/170.4 eV) from

LiFSI salt in the S 2p XPS spectra.[2] This confirms the N atoms would not be bonded with O=S=O moieties to interfere with the identification of C-N species.

Supplementary Fig. 42. Characterization of SEI composition on the cathodes cycled in Li-CO₂ batteries. C 1s, N 1s, and S 2p XPS spectra, and the content comparison of C-N species and SO₄²⁻ for the cathodes after 3, 10, 30, and 50 cycles in the cells using (a) single-salt LiFSI electrolytes.

Furthermore, due to its high sensitivity for carbon detection, C K-edge and N K-edge XANES spectra have been conducted to not only confirm the existence of C-N species but also to clarify the correlation between Li₂CO₃ residuals and C-N species. As shown in Fig. 3e, distinct peaks of C-N species appeared after battery cycling compared to no signals on the pristine cathode. It should be noted that the intensities of Li₂CO₃ residuals gradually decreased with increased peak intensity of C-N species at 404.8 eV in the N K-edge XANES spectra³⁸, further confirming the catalytic effects of C-N species towards Li₂CO₃ decomposition. In particular, the cells in LiNO₃ and LiBF₄ electrolytes with low contents of C-N species show significantly high intensities of Li₂CO₃ residuals peaks (Fig. 3e, yellow and green lines, respectively), suggesting their inferior reversibility of Li₂CO₃ and poor cycling stability. In contrast, the cells in LiFSI electrolytes with a relatively higher content of C-N species show the lowest intensities of Li₂CO₃ residuals (Fig. 3e, red lines), thus superior cycling performance.

Fig. 3 (e) C K-edge and N K-edge XANES spectra for the pristine cathode and cathodes after 3 cycles.

[1] He, H. et al. Investigation of edge-selectively nitrogen-doped metal free graphene for oxygen reduction reaction. *J. Adv. Nanotechnol.* **1**, 5-13 (2020).

[2] Zhang, X.Q. et al. Regulating anions in the solvation sheath of lithium ions for stable lithium metal batteries. *ACS Energy Lett.* **4**, 411-416 (2019).

Comment 5: It is very strange that the bonding between Li^+ ions and anions calculated by the authors shows positive free energy changes. Such electrostatic attractions are naturally favorable as found in their salt forms.

Reply: Thank you for your valuable comment, including revelation of our errors in the signs of ΔG . We have checked and corrected the bonding free energy between Li^+ ions and anions in **Figs. 4b** and **S34**, and amended the relevant discussions in the revised manuscript.

The binding free energy (ΔG_{bind}) was calculated between one Li^+ ion and one anion to examine the bonding strength of different anions with an Li ion (**Fig. S34**).⁴¹ All Li^+ -anion complexes show negative ΔG_{bind} (**Fig. S35**), and a more negative ΔG_{bind} suggests a stronger coordination of anions with Li^+ ion in the solvation shell. Compared to all the other anions, NO_3^- shows the most negative ΔG_{bind} of -0.83 eV, whereas FSI shows the least negative ΔG_{bind} of -0.43 eV (**Fig. 4b**, green columns). The relatively high ΔG_{bind} suggests that FSI preferentially desolvate with Li^+ ion than other anions, and then decomposes on the cathode surface, leading to a high content of C-N species in SEI layer.^{13,43} This reveals that the decomposition reactions depend not only on the solvation structure, but also the coordination capability between different components in inner solvation shells.

Supplementary Fig. 34. Thermodynamic cycle of the charge-transfer reactions involving Li^+ and other electrolyte components: an anion (X) and DMSO, used to calculate the binding energy (ΔG_{bind}) between Li^+ and associated species. The energies for Li vaporisation, Li ionisation, and Li solvation in organic electrolytes were used as in the previous model.^{9,10} A Li solvation energy in DMSO of -5.36 eV was used.¹¹

Supplementary Fig. 35. Binding free energy between Li^+ ion and different anions, as calculated by DFT.

Fig. 4 (b) Left Y-axis: coordination numbers of different anions (yellow) and DMSO solvent (yellow slashes) in inner solvation shells by MD; right Y-axis: Binding free energy ΔG_{bind} between Li^+ and different anions calculated by DFT.

Comment 6: The authors claim the dominance of CIP solvation species in the LiFSI-electrolyte. However, it is generally believed that LiFSI is highly dissociating and it is unlikely to favor CIP in a highly polar solvent like DMSO.

Reply: LiFSI salts have been widely used to construct fluorinated SEI layers on electrodes in battery systems,[1-3] reflecting that FSI⁻ anions in associated state, including contact ion pairs (CIP) and aggregates (AGG), are involved in the formation of SEI.[4] Many studies have reported the existence of CIP for LiFSI in highly polar solvents, such as TEP[5], DOL, DME and DOL/DME mixture.[6-8] This indicates that it is possible to form CIP in LiFSI/DMSO electrolytes.

In this work, we have also provided some experimental and theoretical evidence to prove the formation of CIP in LiFSI/DMSO electrolytes. In the Raman spectrum, it has been widely accepted that a shift of an anion peak to higher wavenumbers in electrolytes indicates that the anion binds with Li^+ ions, and more shift suggests stronger Li^+ -anion interaction.⁴¹ With increasing shift, the peak of S-N-S stretching mode from FSI⁻ can be deconvoluted into two peaks for solvent-separated ion pairs (SSIP, 722 cm^{-1}) and solvent-contact ion pairs (CIP, 731 cm^{-1}) (Table S2). As shown in Fig. 4a, LiNO₃ and LiFSI electrolytes show more CIP (52.5% and 44.65 %, respectively) relative to the other electrolytes, while a smaller proportion of CIP is observed in the others: LiTFSI (13.7 %) and LiBF₄ (12.4 %) (Tables S2, 3). This demonstrates that NO₃⁻ and FSI⁻ have the strongest coordination with Li^+ ions compared to the others, which can promote anion-derived SEI generation on electrodes.

Theoretically, contact ion pairs (CIP) refer to anions in inner solvation shells coordinating with a single Li^+ ion [9-11]. MD simulation has been conducted to reveal the structure of solvation shells in LiFSI-based electrolytes. In RDFs (Fig. S32), the sharp peak at around 2 Å for Li-FSI⁻ coordination indicates the existence of FSI⁻ in inner solvation shells. The average coordination number is 1.42 for LiFSI (Fig. 4b), suggesting a large amount of FSI⁻ coordinated with Li^+ and the existence of CIP. Furthermore, more information was provided by the surrounding environment of Li^+ solvation clusters, which can be classified as solvent-surrounded Li^+ , Li^+ -single anion pair (CIP), and Li^+ -multiple anion cluster by coordination numbers of anions with Li^+ ions of 0, 1, and ≥ 2 (Table S4). In particular, a large preference for Li^+ solvation clusters including anions (50% Li^+ -single anion pairs and 36% Li^+ -multiple anion clusters) is for LiFSI (Fig. 4f), which agrees with the high content of CIP (44.65 %) in the Raman results (Fig. 4a) and distinct ¹⁹F NMR peaks from FSI⁻ corresponding to the Li-FSI⁻ coordination (Fig. S33).

Fig. 4 Theoretical and experimental studies on the Li⁺ solvation structures and electrolyte configurations. (a) Fitted Raman curves of anions for different anion pairs of free anions, SSIP, and CIP, and their ratio comparison. (b) Left Y-axis: coordination numbers of different anions (yellow) and DMSO solvent (yellow slashes) in inner solvation shells by MD; right Y-axis: Binding free energy ΔG_{bind} between Li⁺ and different anions calculated by DFT. MD simulation boxes, representative structures of Li⁺ solvation clusters, and the electrolyte configurations based on the percentages of different Li⁺ solvation clusters for (c) LiNO₃, (d) LiBF₄, (e) LiFSI, and (f) LiTFSI electrolytes.

Supplementary Fig. 32. The radial distribution functions (RDFs) of Li-DMSO/anions and their coordination numbers in 1 M (d) LiFSI electrolytes. The dashed-dotted lines express the coordination numbers.

Supplementary Fig. 33. (a) ^1H NMR, and (b) ^{13}C NMR spectra of pure DMSO solvent and 1 M LiFSI/DMSO electrolytes. (c) ^{19}F NMR spectrum of and 1 M LiFSI/DMSO electrolytes.

Note: An upfield (more negative) shift in ^1H and ^{13}C NMR spectra, respectively, could indicate Li-solvent binding interaction, confirming the existence of Li^+ solvation shells. In addition, the appearance of distinct ^{19}F NMR peaks were observed and attributed to the Li^+ -FSI $^-$ coordination.

Supplementary Table 2. Interaction models between cations and anions, and corresponding binding wavenumbers of vibrations of various anions.

Anions	Free/ cm^{-1}	SSIP/ cm^{-1}	CIP/ cm^{-1}
NO_3^- ¹¹	1039	1045	1051
BF_4^- ¹²	760	763	768
TFSI $^-$ ¹³	736-738	740-742	744-746
FSI $^-$ ^{14,15}	719	722	731

Supplementary Table 3. Fitting results for Raman spectra.

Samples	Content in solvent (%)	Content of anions (%)
---------	------------------------	-----------------------

	Free	Solvation	Free	SSIP	CIP
1M LiNO ₃	57.75	42.25	42.5	0	52.5
1M LiBF ₄	53.25	46.75	0	87.6	12.4
1M LiTFSI	50.31	49.96	35.1	51.2	13.7
1M LiFSI	52.7	47.3	0	55.35	44.65

Supplementary Table 4. The percentages of different Li⁺ solvation clusters in 1 M LiNO₃, LiBF₄, LiTFSI, and LiFSI electrolytes, calculated by MD.

Li ⁺ solvation cluster	NO ₃ ⁻	BF ₄ ⁻	TFSI ⁻	FSI ⁻
Solvent-surrounded Li ⁺	8%	23%	56%	14%
Li ⁺ -single anion pair	37%	33%	38%	50%
Li ⁺ -multiple anion cluster	55%	43%	6%	36%

- [1] Li, T. et al. Fluorinated solid-electrolyte interphase in high-voltage lithium metal batteries. *Joule* **3**, 2647-2661 (2019).
- [2] Zhao, Y. et al. Fluorinated ether electrolyte with controlled solvation structure for high voltage lithium metal batteries. *Nat. Commun.* **13**, 2575 (2022).
- [3] Cao, X. et al. Monolithic solid-electrolyte interphases formed in fluorinated orthoformate-based electrolytes minimize Li depletion and pulverization. *Nat. Energy* **4**, 796-805 (2019).
- [4] Chang, Z. et al. Beyond the concentrated electrolyte: further depleting solvent molecules within a Li⁺ solvation sheath to stabilize high-energy-density lithium metal batteries. *Energy Environ. Sci.* **13**, 4122-4131 (2020).
- [5] Zeng, Z. et al. Non-flammable electrolytes with high salt-to-solvent ratios for Li-ion and Li-metal batteries. *Nat. Energy* **3**, 674-681 (2018).
- [6] Jiang, K.S. et al. Probing the functionality of LiFSI structural derivatives as additives for Li metal anodes. *ACS Energy Lett.* **7**, 3378-3385 (2022).
- [7] Zhao, Y. et al. Electrolyte engineering via ether solvent fluorination for developing stable non-aqueous lithium metal batteries. *Nat. Commun.* **14**, 299 (2023).
- [8] Holoubek, J. et al. Tailoring electrolyte solvation for Li metal batteries cycled at ultra-low temperature. *Nat. Energy* **6**, 303-313 (2021).
- [9] Jiao, S. et al. Stable cycling of high-voltage lithium metal batteries in ether electrolytes. *Nat. Energy* **3**, 739-746 (2018).
- [10] Qiao, Y. et al. Li₂CO₃-free Li-O₂/CO₂ battery with peroxide discharge product. *Energy Environ. Sci.* **11**, pp.1211-1217 (2018).
- [11] Zhang, Y. et al. Solvent molecule cooperation enhancing lithium metal battery performance at both electrodes. *Angew. Chem. Int. Ed.* **59**, 7797-7802 (2020).

Reply to Reviewer #2

Overall comment: This paper uncovers the significant contribution of C-N species within the SEI layer. These species play a crucial role by offering adsorption sites and facilitating rapid charge transfer to $^*CO_2^{2-}$ and Li_2CO_3 , thereby effectively controlling the reversibility of Li_2CO_3 throughout battery cycling. Furthermore, the authors have successfully engineered a C-N enriched SEI layer within a dual salts electrolyte, resulting in sustained long-term cycling performance. This achievement offers fresh perspectives on SEI structure, electrolyte formulations, and the design of catalytic active sites in Li-CO₂ batteries. However, it is still not clear to me the dynamics of the key redox reaction of $3CO_2 + 4Li + 3e^- \leftrightarrow 2Li_2CO_3 + C$ and effects of lithium salt LiFSI, more evidence and explanation would be recommended. Overall, I would like to reconsider its publication if the following concerns can be addressed.

Reply: We value your comments and suggestions, which have been very helpful for improving the quality of our manuscript. We have carefully revised the paper according to these suggestions and have done our best to answer your questions. We sincerely hope that the manuscript will meet with your approval after revision.

Comment 1: As the $^*CO_2^{2-}$ is a key intermediate radical from CO_2 to Li_2CO_3 and authors do some FTIR work to demonstrate the formation of Li_2CO_3 in Fig. S11, I wonder if it's more straightforward to confirm the existence of $^*CO_2^{2-}$ during reactions by powerful FTIR technique.

Reply: Thank you for your insightful suggestion. It is generally known that CO_2 undergoes several possible intermediate processes and eventually gets converted to Li_2CO_3 in Li-CO₂ batteries. These are demonstrated through equations S1-4. First, CO_2 experiences a one-electron reduction to be reduced to $C_2O_4^{2-}$ (eq S1). The intermediate $C_2O_4^{2-}$ (eq S2, 3) produces $^*CO_2^{2-}$ radicals by a one-electron reaction and then react with $^*CO_2^{2-}$ to form Li_2CO_3 (eq S4). [1-4]

We have conducted powerful *in-situ* surface-enhanced Raman spectroscopy (SERS) to investigate the discharge mechanism in Li-CO₂ batteries and monitor the variation trends of discharge species. In order to capture information about intermediates during discharging, CO_2 reduction has been decelerated by reducing both the Li-ion (50 mM) and CO_2 ($CO_2:Ar = 1:5$, v/v) concentrations in the electrolyte. After controlling CO_2 partial pressure and Li-ion concentration, *in-situ* Raman spectra were collected on a gold substrate with constant voltage of 2.2 V to further increase the signals of intermediate and discharge products.

Supplementary Fig. 1. Observation of intermediates during the CO₂ reduction process, via *in-situ* surface-enhanced Raman spectroscopy (SERS).

Note: Although it is hard to directly detect the existence of *CO₂²⁻ due to its instability, especially during battery cycling [5,6], C₂O₄²⁻ could be detected as the intermediate at the beginning of discharge. This is revealed in Fig. S1, where the newly-emerged peaks at 876 and 907 cm⁻¹ can be assigned to C-C stretching modes, and the pair of peaks at 1492 and 1653 cm⁻¹ can be attributed to the C-O stretching modes in C₂O₄²⁻. Subsequently, the C₂O₄²⁻ intermediate gradually disappeared. This was accompanied by the appearance of peaks of 1319 and 1587 cm⁻¹ for D and G band, respectively, and the Li₂CO₃ (1089 cm⁻¹), a result consistent with the above reaction pathway of CO₂ reduction (eq S1-4).

- [1] Hou, Y. et al. Mo₂C/CNT: An Efficient Catalyst for Rechargeable Li-CO₂ Batteries. *Adv. Funct. Mater.* **27**, 1700564 (2017).
- [2] Qiao, Y. et al. Li-CO₂ Electrochemistry: A New Strategy for CO₂ Fixation and Energy Storage. *Joule* **1**, 359-370 (2017).
- [3] Mu, X. et al. Li-CO₂ and Na-CO₂ Batteries: Toward Greener and Sustainable Electrical Energy Storage. *Adv. Mater.* **32**, 1903790 (2020).
- [4] Jiao, Y. et al. Recent progress and prospects of Li-CO₂ batteries: Mechanisms, catalysts and electrolytes. *Energy Storage Mater.* **34**, 148-170 (2021).
- [5] Kafafi, Z.H. et al. Carbon dioxide activation by lithium metal. 1. Infrared spectra of lithium carbon dioxide (Li⁺ CO₂⁻), lithium oxalate (Li⁺ C₂O₄⁻), and lithium carbon dioxide (Li₂²⁺ CO₂²⁻) in inert-gas matrices. *J. Am. Chem. Soc.* **105**, 3886-3893 (1983).
- [6] Kafafi, Z.H. et al. Carbon dioxide activation by alkali metals. 2. Infrared spectra of M⁺ CO₂⁻ and M₂²⁺ CO₂²⁻ in argon and nitrogen matrixes. *Inorg. Chem.* **23**, 177-183 (1984).

Comment 2: Given that the authors have employed multiscale characterizations, including FTIR, Raman, TEM, and XPS, to substantiate the presence of Li₂CO₃, it raises the question of whether the stability of the cathode materials was considered during exposure to air or during the sample transfer process after cycles.

Reply: Thank you for your valuable comment. The insoluble discharge product Li_2CO_3 is thermodynamically stable (under 600°C) and cannot be decomposed easily during the sample transfer process. To avoid the sample pollution from exposure to air, the Li- CO_2 cells were disassembled in an argon-filled glovebox, and then the cycled cathodes were directly transferred to the X-ray photoelectron spectroscopy connected with the argon-filled glovebox for XPS measurement. So the samples had no chance of being exposed to air.

In order to eliminate the possible sample pollution, *in-situ* Raman evidence was provided for the Li- CO_2 cells to prove the formation/decomposition of Li_2CO_3 during battery cycling (Fig. S19). Taking LiNO_3 and LiFSI electrolytes as examples, the peak of Li_2CO_3 can be clearly observed to emerge during discharge and gradually disappear during charging (Fig. S20), suggesting the excellent reversibility of the cathodes. Furthermore, signals of DMSO_2 species (green *) were detected, which suggests decomposition of DMSO solvent due to the attack of reduced $\text{CO}_2^{\cdot-}$ radicals during cycling. Compared to the strong signals of DMSO_2 species in LiNO_3 cells, the signals in LiFSI cells gradually disappeared after being fully recharged, verifying that solvent degradation could be suppressed to some extent, in accord with its superior cycling stability. We have added the relevant discussion in the revised manuscript.

Supplementary Fig. 19. Schematic of *in-situ* Raman cell.

Note: For *in-situ* Raman test, CR2032-type coin cells (a hole on the cathode side) were assembled in an Ar-filled glove box with air electrodes and lithium chip anodes separated by a glass fiber separator (Whatman, diameter: 19 mm). Solutions of 1 M $\text{LiNO}_3/\text{DMSO}$ and 1 M LiFSI/DMSO were used as electrolytes. The as-prepared coin cells were sealed in *in-situ* Raman cell and purged with CO_2 gas during the test process.

Supplementary Fig. 20. In-situ Raman evidence for the Li-CO₂ cells with the formation/decomposition of Li₂CO₃ during discharge/charge. The *in-situ* Raman spectra collected at specific points (marked at voltage profile) during battery running at a constant current of 100 μ A with a cut-off capacity of 30 μ Ah for (a) LiNO₃ cells and (b) LiFSI cells. The peaks assigned to dimethyl sulfone (DMSO₂) species were observed during cycling. Corresponding voltage profiles are listed above each group of the *in-situ* Raman spectra.

Comment 3: Based on the above two comments, I wonder if *in situ* FTIR or Raman experiments can be done to prove the dynamics, for example, the existence of $^{*}\text{CO}_2^{2-}$ and Li₂CO₃, during cycling.

Reply: Thank you for your insightful suggestion. *In-situ* surface-enhanced Raman spectroscopy (SERS) was provided above (Comment 1, Reviewer #2) to investigate the discharge mechanism in Li-CO₂ batteries and monitor the variation trends of discharge species (Li₂CO₃) on a gold substrate.

Supplementary Fig. 1. Observation of intermediates during the CO₂ reduction process, via *in-situ* surface-enhanced Raman spectroscopy (SERS).

Note: Although it is hard to directly detect the existence of $\cdot\text{CO}_2^{2-}$ due to its instability, especially during battery cycling [5,6], $\text{C}_2\text{O}_4^{2-}$ could be detected as the intermediate at the beginning of discharge. This is revealed in Fig. S1, where the newly-emerged peaks at 876 and 907 cm^{-1} can be assigned to C-C stretching modes, and the pair of peaks at 1492 and 1653 cm^{-1} can be attributed to the C-O stretching modes in $\text{C}_2\text{O}_4^{2-}$. Subsequently, the $\text{C}_2\text{O}_4^{2-}$ intermediate gradually disappeared. This was accompanied by the appearance of peaks of 1319 and 1587 cm^{-1} for D and G band, respectively, and the Li_2CO_3 (1089 cm^{-1}), a result consistent with the above reaction pathway of CO₂ reduction (eq S1-4).

We have also provided *in-situ* Raman evidence for the Li-CO₂ cells to prove the formation/decomposition of Li_2CO_3 during battery cycling (Comment 2, Reviewer #2).

Taking LiNO_3 and LiFSI electrolytes as examples, the peak of Li_2CO_3 can be clearly observed to emerge during discharge and gradually disappear during charging (Fig. S20), suggesting the excellent reversibility of the cathodes. Furthermore, signals of DMSO₂ species (green *) were detected, which suggests decomposition of DMSO solvent due to the attack of reduced $\cdot\text{CO}_2^{2-}$ radicals during cycling. Compared to the strong signals of DMSO₂ species in LiNO_3 cells, the signals in LiFSI cells gradually disappeared after being fully recharged, verifying that solvent degradation could be suppressed to some extent, in accord with its superior cycling stability.

Supplementary Fig. 20. In-situ Raman evidence for the Li-CO₂ cells with the formation/decomposition of Li₂CO₃ during discharge/charge. The *in-situ* Raman spectra collected at specific points (marked at voltage profile) during battery running at a constant current of 100 μ A with a cut-off capacity of 30 μ Ah for (a) LiNO₃ cells and (b) LiFSI cells. The peaks assigned to dimethyl sulfone (DMSO₂) species were observed during cycling. Corresponding voltage profiles are listed above each group of the *in-situ* Raman spectra.

Comment 4: How to understand the difference in EIS for all four salts used in the paper? It appears that the EIS results for LiNO₃, LiTFSI, and LiBF₄ do not exhibit significant distinctions. As the distinct EIS data from LiFSI is used to explain the superior performance, it's recommended to clarify how the variations in the performance of other salts are explained despite the apparent similarities in their EIS data?

Reply: Thank you for your kind comments. **Figs. 2b** and **S6** show electrochemical impedance spectra (EIS) of Li-CO₂ full-cells in various single-salt electrolytes. After the first discharge, all the cells display a dramatic increase in the battery impedance due to the precipitation of insulating discharge product of Li₂CO₃. After subsequent recharge, full recovery of the impedance spectrum could only be observed for the LiFSI cell (**Fig. 2b**) after a discharge-charge cycle compared with a partial recovery for the others (**Fig. S6**), suggesting the complete decomposition of deposited Li₂CO₃ upon recharging in LiFSI cells.

Fig. 2 (b) EIS spectra of the LiNO₃ (top) and LiFSI (bottom) cells before discharge, after the first discharge, and after recharge.

Supplementary Fig. 6. EIS spectra of the (a) LiBF₄, and (b) LiTFSI cells before discharge, after the first discharge, and after recharge.

It is generally known that the poor cyclability of Li-CO₂ cells is mainly caused by the gradual accumulation of the undecomposed Li₂CO₃ on the cathode surface with cycling. To clarify the influence of these single-salt electrolytes on cycling performance, we have added EIS measurement of Li-CO₂ full-cells at different cycles (**Fig. S7** in the revised manuscript). **The increase in impedance with cycling follows the trend of LiFSI < LiTFSI < LiBF₄ < LiNO₃, consistent with their cycling profiles (cycle life decreases in the order of LiFSI > LiBF₄ > LiTFSI > LiNO₃).** As expected, the impedances of LiFSI cells were maintained to be the lowest, while those of LiNO₃ cells vastly increased after just 20 cycles.

Supplementary Fig. 7. EIS spectra of the Li-CO₂ full-cells in the (a) LiNO₃, (b) LiBF₄, (c) LiTFSI, and (d) LiFSI electrolytes (fully charged state) with cycling (before cycling, and after 5 cycles, 10 cycles, 20 cycles, 30 cycles, 40 cycles, 50 cycles.).

Comment 5: Please enhance the image quality in Fig. S10, as the current quality does not convincingly confirm the presence of Li₂CO₃. The existing image lacks the clarity needed to discern any lattice structure, and it is challenging to identify the specific features that the authors intend to emphasize in the TEM images. Additionally, the yellow circles drawn by the authors seem to obscure valuable information from the FFT, and I am uncertain whether I can concur with the authors' conclusions regarding this matter.

Reply: Thank you for your valuable suggestions. We have removed the yellow circles in Fig. S10 (Fig. S14 in the revised manuscript) and improved the image quality.

Supplementary Fig. 14. High-resolution transmission electron microscope (HRTEM) images and corresponding inset fast Fourier transform patterns (FFT) of the cathodes discharged in the cells using 1 M LiNO₃, LiBF₄, LiTFSI, and LiFSI electrolytes.

To further confirm the formation of Li₂CO₃ after discharge, the scanning transmission electron microscopy (STEM) images and the selected area electron diffraction (SAED) pattern for the cathode discharged in LiNO₃ cells were provided in Fig. S15.

Supplementary Fig. 15. (a-b) Scanning transmission electron microscopy (STEM) images in different scale and (c) the selected area electron diffraction (SAED) pattern of the cathodes discharged in the cells using 1 M LiNO₃ electrolytes.

Note: The discharge products were observed to be grown on rGO sheets (Fig. S15a). Except for the (002) lattice plane of graphene, other diffraction rings can be confirmed as (112), (220), (511), (330), and (621) lattice planes of Li₂CO₃ in SAED (Fig. S15c).

Comment 6: What's the primary distinction between lithium salts LiFSI and LiTFSI? It's worth noting that LiTFSI can also offer enhanced anion coordination and contains a similar nitrogen element to LiFSI. This raises questions about why a single LiFSI salt is more effective at facilitating the formation of C-N species within the SEI, and this aspect should be clarified for better understanding.

Reply: Thank you for your insightful comment. LiTFSI has a similar nitrogen element to LiFSI,

however, LiTFSI exhibits a much weaker coordination ability with Li⁺ ions than that of LiFSI in electrolytes due to its relatively larger size. MD simulation results show that the average coordination number of TFSI with Li⁺ is 0.49, which is much less than that (1.42) for FSI (Fig. S32). The surrounding environment of Li⁺ solvation clusters provide more detailed information, which can be classified as solvent-surrounded Li⁺, Li⁺-single anion pair, and Li⁺-multiple anion cluster by coordination numbers of anions with Li⁺ ions of 0, 1, and ≥ 2. A large preference for Li⁺ solvation clusters including anions (50% Li⁺-single anion pairs and 36% Li⁺-multiple anion clusters) is for LiFSI (Fig. 4e), whereas solvent-surrounded Li⁺ clusters dominate for the LiTFSI electrolyte (Fig. 4f). In addition, representative solvated (Li-FSI-DMSO₄) clusters in LiFSI electrolytes present a slightly lower LUMO energy value (-1.09 eV) than that (-0.93 eV) for (Li-TFSI-DMSO₄) clusters in LiTFSI electrolytes (Fig. 5f). This suggests that FSI⁻ exhibits higher possibility of accepting electrons to preferentially be reduced to generate anion-derived SEI on electrodes than does TFSI⁻. Therefore, FSI⁻ not only has a stronger coordination with Li⁺ ions but also displays a lower LUMO energy level compared to TFSI⁻, both of which can effectively facilitate the formation of C-N species in the SEI on electrodes.

Supplementary Fig. 32. The radial distribution functions (RDFs) of Li-DMSO/anions and their coordination numbers in 1 M (c) LiTFSI and (d) LiFSI electrolytes. The dashed-dotted lines express the coordination numbers.

Fig. 4 MD simulation boxes, representative structures of Li⁺ solvation clusters, and the electrolyte configurations based on the percentages of different Li⁺ solvation clusters for (e) LiFSI, and (f) LiTFSI electrolytes.

Fig. 5 (f) LUMO energy values for representative Li⁺ solvation clusters in single-salt and dual-salt

electrolytes.

Comment 7: As LiTFSI can also provide more anion coordination and contains the similar N element with LiFSI. I wonder why single LiFSI salt can promote the formation of C-N in the SEI should be clearer.

Reply: Thank you for your insightful comment. Detailed explanation of why a single LiFSI salt is more effective at facilitating the formation of C-N species within the SEI than LiTFSI was provided above (Comment 6, Reviewer #2).

LiTFSI has a similar nitrogen element to LiFSI, however, LiTFSI exhibits a much weaker coordination ability with Li^+ ions than that of LiFSI in electrolytes due to its relatively larger size. MD simulation results show that the average coordination number of TFSI $^-$ with Li^+ is 0.49, which is much less than that (1.42) for FSI $^-$ (Fig. S32). The surrounding environment of Li^+ solvation clusters provide more detailed information, which can be classified as solvent-surrounded Li^+ , Li^+ -single anion pair, and Li^+ -multiple anion cluster by coordination numbers of anions with Li^+ ions of 0, 1, and ≥ 2 . A large preference for Li^+ solvation clusters including anions (50% Li^+ -single anion pairs and 36% Li^+ -multiple anion clusters) is for LiFSI (Fig. 4e), whereas solvent-surrounded Li^+ clusters dominate for the LiTFSI electrolyte (Fig. 4f). In addition, representative solvated (Li-FSI-DMSO $_4$) clusters in LiFSI electrolytes present a slightly lower LUMO energy value (-1.09 eV) than that (-0.93 eV) for (Li-TFSI-DMSO $_4$) clusters in LiTFSI electrolytes (Fig. 5f). This suggests that FSI $^-$ exhibits higher possibility of accepting electrons to preferentially be reduced to generate anion-derived SEI on electrodes than does TFSI $^-$. Therefore, FSI $^-$ not only has a stronger coordination with Li^+ ions but also displays a lower LUMO energy level compared to TFSI $^-$, both of which can effectively facilitate the formation of C-N species in the SEI on electrodes.

Supplementary Fig. 32. The radial distribution functions (RDFs) of Li-DMSO/anions and their coordination numbers in 1 M (c) LiTFSI and (d) LiFSI electrolytes. The dashed-dotted lines express the coordination numbers.

Fig. 4 MD simulation boxes, representative structures of Li^+ solvation clusters, and the electrolyte configurations based on the percentages of different Li^+ solvation clusters for (e) LiFSI, and (f) LiTFSI electrolytes.

Fig. 5 (f) LUMO energy values for representative Li^+ solvation clusters in single-salt and dual-salt electrolytes.

Comment 8: What is the role or function of the Ru nanoparticles? The introduction of Ru nanoparticles in the paper appears somewhat abrupt. It would be beneficial to clarify the purpose of these nanoparticles and provide information on the electrode's performance in the absence of Ru nanoparticles.

Reply: Thank you for your kind comments. Ru nanoparticles have been widely considered to have excellent catalytic activity towards the CO_2 evolution during the charging process in $\text{Li}-\text{CO}_2$ batteries [1-4], and its employment in this work aims to confirm that the effects of C-N species towards battery cycling is applicable with the participation of a metal catalyst. As shown in **Fig. S11**, the cycle life of the cells using an Ru catalyst follow the same tendency of $\text{LiFSI} > \text{LiTFSI} > \text{LiBF}_4 > \text{LiNO}_3$ as that using a metal-free catalyst, rGO. At a current density of 0.3 A g^{-1} , the cell using LiFSI electrolytes maintains the set cut-off capacity after 630 hours, whereas the cell using LiNO_3 electrolytes exhibits capacity decay after 250 hours.

Supplementary Fig. 11. Cycling performance of $\text{Li}-\text{CO}_2$ batteries with a ruthenium (Ru) catalyst at a current density of 0.3 A g^{-1} with a cut-off specific capacity of 500 mA h g^{-1} in 1 M LiNO_3 , LiBF_4 , LiTFSI , and LiFSI electrolytes.

We have also added the cycling performance of $\text{Li}-\text{CO}_2$ cells in the absence of catalysts (rGO and Ru nanoparticles) in the revised manuscript to further verify the catalytic role of C-N species, which

improves the cycling performance of the Li-CO₂ batteries. As shown in **Fig. S10**, the cells with carbon papers as cathodes exhibited the same trend of cycle life as that with Ru or rGO catalysts, which trends in the order of LiFSI > LiTFSI > LiBF₄ > LiNO₃. It should be noted that these cells show poor cycling performance and severe electrolyte decomposition upon the first two cycles, even under a small current density of 50 mA g⁻¹. Thus, the application of catalysts in Li-CO₂ batteries is necessary to lower the overpotential and improve the battery cycling stability.

Supplementary Fig. 10. Cycling performance of Li-CO₂ batteries without catalysts at a current density of 50 mA g⁻¹ with a cut-off specific capacity of 500 mA h g⁻¹ in 1 M LiNO₃, LiBF₄, LiTFSI, and LiFSI electrolytes.

- [1] Sun, X. et al. Binuclear Cu complex catalysis enabling Li-CO₂ battery with a high discharge voltage above 3.0 V. *Nat. Commun.* **14**, 536 (2023).
- [2] Wang, D. et al. A low-charge-overpotential lithium-CO₂ cell based on a binary molten salt electrolyte. *Energy Environ. Sci.* **14**, 4107-4114 (2021).
- [3] Mu, X. et al. Li-CO₂ and Na-CO₂ batteries: toward greener and sustainable electrical energy storage. *Adv. Mater.* **32**, 1903790 (2020).
- [4] Sarkar, A. et al. 2023. Recent Advances in Rechargeable Metal-CO₂ Batteries with Nonaqueous Electrolytes. *Chem. Rev.* **123**, 9497-9564 (2023).

Comment 9: Will the SEI continue to undergo cycles of collapse and reconstruction after multiple charge-discharge cycles? How can the effective function of C-N species be ensured throughout these cycles?

Reply: Thank you for your valuable comments. We agree that SEI may experience collapse and reconstruction during cycling, which is a challenging issue to be addressed in battery field.[1-4]

To reveal the variations of C-N species in SEI layer, we have compared the XPS analysis results for the cathodes after 3, 10, 30, and 50 cycles in cells using single-salt LiFSI and 0.25 M LiNO₃/0.75 M LiFSI electrolytes, as shown in **Figs. S42** and **43** in the revised manuscript. The N 1s XPS spectra show there was no obvious change in the content of C-N species with charge/discharge cycling, suggesting that C-N species remained on the surface of cathodes even after prolonged cycling, and that contributing to the

excellent cycling performance of the batteries. Compared to single-salt LiFSI cells (about 0.8% C-N species), 0.25 M LiNO₃/0.75 M LiFSI cells present a relatively higher content of C-N species (1.74%). Accordingly, C 1s XPS spectra show that no Li₂CO₃ residuals were detected in 0.25 M LiNO₃/0.75 M LiFSI cells even after 50 cycles, whereas LiFSI cells display an obvious peak of Li₂CO₃ at around 290.6 eV (★ symbol) after 30 cycles, in agreement with their cycling profiles. These results confirm the catalytic effect of C-N species towards promoting Li₂CO₃ decomposition and prolonging battery cycling.

Supplementary Fig. 42. Characterization of SEI composition on the cathodes cycled in Li-CO₂ batteries. C 1s, N 1s, and S 2p XPS spectra, and Comparison of C-N species and SO₄²⁻ content for the cathodes after 3, 10, 30, and 50 cycles in the cells using (a) single-salt LiFSI and (b) 0.25 M LiNO₃/0.75 M LiFSI electrolytes.

Supplementary Fig. 43. XPS survey spectra of the cathodes after 3, 10, 30, and 50 cycles in the cells using (a) single-salt LiFSI and (b) 0.25 M LiNO₃/0.75 M LiFSI electrolytes.

- [1] Cao, D., Tan, C. and Chen, Y. Oxidative decomposition mechanisms of lithium carbonate on carbon substrates in lithium battery chemistries. *Nat. Commun.* **13**, 4908 (2022).
- [2] Wang, L. et al. Identifying the components of the solid–electrolyte interphase in Li-ion batteries. *Nat. Chem.* **11**, 789-796 (2019).
- [3] Li, T. et al. Fluorinated solid-electrolyte interphase in high-voltage lithium metal batteries. *Joule* **3**, 2647-2661 (2019).
- [4] Guo, W. et al. Artificial dual solid-electrolyte interfaces based on in situ organothiol transformation in lithium sulfur battery. *Nat. Commun.* **12**, 3031 (2021).

Comment 10: As the pristine cathode contains a wide variety of C-SO₃ species according to XPS in Fig S13, how can authors distinguish Li₂CO₃ and C-SO₃ as they share the similar binding energy?

Reply: Thank you for your valuable comment. We have provided the high resolution XPS C 1s spectra of the cathodes before and after discharging in various electrolytes in **Fig. S13** (**Fig. S18** in the revised manuscript) to detect the discharge products. The pristine cathode shows the non-oxygenated ring carbon (C-C) and C-O peaks from rGO (284.5 eV and 285 eV, respectively), and C-SO₃ (289.5 eV), CF₂ (~293 eV) and CF₃ (~294 eV) peaks from Nafion binder.[1-3] After discharge, the emerged peak of Li₂CO₃ at 290.6 eV (red dash line) can be easily distinguished due to the obvious difference in binding energy of C-SO₃ (289.5 eV, peak dash line) species, verifying its formation as discharge products.[4-6]

Supplementary Fig. 18. C 1s XPS spectra of the pristine cathode and the cathodes discharged in the cells using 1 M LiNO₃, LiBF₄, LiTFSI, and LiFSI electrolytes. (The peaks at 284.5 eV and 285 eV can be ascribed to non-oxygenated ring carbon (C-C) and the presence of oxygen in rGO (C-O), respectively; C-O/C=O: 287.7 eV, CF₂: ~293 eV).¹⁻³

- [1] Li, Y. et al. Highly surface-wrinkled and N-doped CNTs anchored on metal wire: A novel fiber-shaped cathode toward high-performance flexible Li–CO₂ batteries. *Adv. Funct. Mater.* **29**, 1808117 (2019).
- [2] Zhang, J. et al. Rechargeable Li-CO₂ Batteries with Graphdiyne as Efficient Metal-Free Cathode Catalysts. *Adv. Funct. Mater.* **31**, 2101423 (2021).
- [3] Qiao, Y. et al. 3D-Printed Graphene Oxide Framework with Thermal Shock Synthesized Nanoparticles for Li-CO₂ Batteries. *Adv. Funct. Mater.* **28**, 1805899 (2018).
- [4] Li, X. et al. Covalent-organic-framework-based Li-CO₂ batteries. *Adv. Mater.* **31**, 1905879 (2019).
- [5] Ye, F. et al. Topological defect - rich carbon as a metal-free cathode catalyst for high-performance Li-CO₂ batteries. *Adv. Energy Mater.* **11**, 2101390 (2021).
- [6] Wang, Y. et al. Decreasing the overpotential of aprotic Li-CO₂ batteries with the in-plane alloy structure in ultrathin 2D Ru-based nanosheets. *Adv. Funct. Mater.* **32**, 2202737 (2022).

Reply to Reviewer #3

Overall comment: *This manuscript by Guo et al. investigated the effect of C–N species in the formed solid electrolyte interphase (SEI) on the generation and decomposition of Li₂CO₃ during discharge and charge processes of Li-CO₂ battery. Theoretical analysis and experimental tests showed that C–N species enable strong adsorption sites and promote charge transfer from interface to *CO₂²⁻ during discharge, and from Li₂CO₃ to C-N species during charge. The results can provide useful information on tuning the component of SEI for improved cycling stability of Li-CO₂ batteries. Before possible consideration of publication in Nat. Commun., the authors need to address the following issues.*

Reply: Thank you very much for your insightful comments and positive opinion on our manuscript. We have carefully revised the manuscript according to your suggestions.

Comment 1: *Why did the authors choose to use rGO with poor catalytic performance as a matrix to investigate the role of the SEI components?*

Reply: We choose rGO as a catalyst due to the following reasons. Firstly, this work mainly aims to investigate the influence of SEI components on the reversibility of Li₂CO₃, thereby governing battery performance. To avoid metal catalyst masking the role of SEI components in this work, we choose to load rGO on carbon papers as matrix.

Secondly, rGO with relatively poor catalytic performance could largely reduce the contribution of catalysts to the Li₂CO₃ decomposition, which will help to justify the role of SEI composition to the Li₂CO₃ decomposition.

In this work, to further test the effect of various electrolytes on battery performance, Li-CO₂ cells were also assembled by using carbon papers as matrix without catalysts (Fig. S10) or using cathodes with Ru catalysts (Fig. S11), respectively, which exhibited the same tendency of cycling performance as that using rGO as matrix. As shown in Fig. S10, the cells using carbon paper as matrix exhibited poor cycling performance and severe electrolyte decomposition upon the first two cycles, even under a small current density of 50 mA g⁻¹. By using Ru as catalysts, the cells in LiFSI-based electrolytes maintain the cut-off capacity after 630 hours at 0.3 A g⁻¹, whereas the cells in LiNO₃-based electrolytes exhibit capacity decay

after 250 hours (Fig. S11).

Supplementary Fig. 10. Cycling performance of Li-CO₂ batteries without catalysts at a current density of 50 mA g⁻¹ with a cut-off specific capacity of 500 mA h g⁻¹ in 1 M LiNO₃, LiBF₄, LiTFSI, and LiFSI electrolytes.

Supplementary Fig. 11. Cycling performance of Li-CO₂ batteries with a ruthenium (Ru) catalyst at a current density of 0.3 A g⁻¹ with a cut-off specific capacity of 500 mA h g⁻¹ in 1 M LiNO₃, LiBF₄, LiTFSI, and LiFSI electrolytes.

Comment 2: There are some format issues that can be corrected. For example, in the caption of X axis in Fig. 2e, “Cycles” can be changed as “Cycle”; the first letter of the words except the first word can be lower-case; among others.

Reply: Thank you for your valuable suggestion. The captions in Figures in the manuscript and

supplementary information have been carefully checked. We have revised the caption of X axis “Cycles” to “Cycle” in Fig.2e, and “Specific Capacity” to “Specific capacity” in Figs.2c and d in the revised manuscript.

Fig. 2 Electrochemical performance of Li-CO₂ cells in various electrolytes. (a) Long-term cycling performance at 0.1 A g⁻¹ (top) and 0.2 A g⁻¹ (bottom) with a cut-off capacity of 500 mAh g⁻¹. **(b)** EIS spectra of the LiNO₃ (top) and LiFSI (bottom) cells before discharge, after the first discharge, and after recharge. **(c)** Full-discharge curves at 0.1 A g⁻¹. **(d)** Discharge-charge curves of the LiFSI cell taken at variant current densities. **(e)** Overpotential comparison of the various electrolytes at different current densities.

Comment 3: Why did the overpotential jump at 2 A g⁻¹ especially for LiTFSI as shown in Fig. 2e?

Reply: Thank you for your kind comment. High polarization and poor rate capability are practical issues for Li-CO₂ batteries, which is mainly attributed to sluggish kinetics of the decomposition of carbonate products during the charging process. Most of recently reported Li-CO₂ batteries displayed a similar overpotential jump when increasing current densities to 2 A g⁻¹. [1-4] Such huge increases in current densities, jumping from 1 to 2 A g⁻¹, will lead to the accelerated generation rate of Li₂CO₃ and corresponding agglomeration and increased particle size of Li₂CO₃, resulting in overpotential jump at 2 A g⁻¹.

To reveal the morphology of discharged products, the Li-CO₂ cells were disassembled after discharge at different current densities of 0.1 and 2 A g⁻¹ (Fig. S9). Compared to the cathodes discharged at 0.1 A g⁻¹, an increased amount and severely agglomerated of Li₂CO₃ particles was observed at 2 A g⁻¹. This

would result in difficulties in decomposition of discharge products and increased overpotential.[5-7] In addition, the surface of the cathodes in LiTFSI electrolytes showed larger particle size of Li_2CO_3 than that in LiFSI electrolytes, which can severely hinder its decomposition reaction kinetics, leading to a higher charge voltage.[8-11] This explains the large overpotential in LiTFSI electrolytes (**Fig. 2e**).

Fig. 2 (e) Overpotential comparison of the various electrolytes at different current densities.

Supplementary Fig. 9. Scanning electron microscope (SEM) images of the pristine cathode and cathodes discharged in the cells using 1 M LiTFSI and LiFSI electrolytes at current densities of 0.1 and 2 A g⁻¹.

- [1] Li, X. et al. Covalent-organic-framework-based Li-CO₂ batteries. *Adv. Mater.* **31**, 1905879 (2019).
- [2] Ye, F. et al. Topological defect-rich carbon as a metal-free cathode catalyst for high-performance Li-CO₂ batteries. *Adv. Energy Mater.* **11**, 2101390 (2021).
- [3] Wang, Y. et al. Decreasing the overpotential of aprotic Li-CO₂ batteries with the in-plane alloy structure in ultrathin 2D Ru-based nanosheets. *Adv. Funct. Mater.* **32**, 2202737 (2022).
- [4] Zhang, K. et al. Boosting cycling stability and rate capability of Li-CO₂ batteries via synergistic photoelectric effect and plasmonic interaction. *Angew. Chem. Int. Ed.* **134**, 202201718 (2022).
- [5] Hou, Y. et al. Mo₂C/CNT: An Efficient Catalyst for Rechargeable Li-CO₂ Batteries. *Adv. Funct.*

Mater. **27**, 1700564 (2017).

[6] Qiao, Y. et al. Li-CO₂ Electrochemistry: A New Strategy for CO₂ Fixation and Energy Storage. *Joule* **1**, 359-370 (2017).

[7] Pipes, R. et al. Phenyl disulfide additive for solution-mediated carbon dioxide utilization in Li-CO₂ batteries. *Adv. Energy Mater.* **9**, 1900453 (2019).

[8] Song, L. et al. An ultra-long life, high-performance, flexible Li-CO₂ battery based on multifunctional carbon electrocatalysts. *Nano Energy* **71**, 104595 (2020).

[9] Chen, B. et al. Engineering the active sites of graphene catalyst: From CO₂ activation to activate Li-CO₂ batteries. *ACS Nano* **15**, 9841-9850 (2021).

[10] Cheng, J. et al. Homogenizing Li₂CO₃ nucleation and growth through high-density single-atomic Ru loading toward reversible Li-CO₂ reaction. *ACS Appl. Mater. Interfaces.* **14**, 18561-18569 (2022).

[11] Wang, Y. et al. 2022. Decreasing the overpotential of aprotic Li-CO₂ batteries with the in-plane alloy structure in ultrathin 2D Ru-based nanosheets. *Adv. Funct. Mater.* **32**, 2202737 (2022).

Comment 4: For the sake of comparison, some spectroscopy measurements may also be conducted prior to battery cycle.

Reply: Thank you for your valuable suggestion. To confirm the correlation between the reversibility of Li₂CO₃ and battery cycling performance, we have provided Raman, FTIR, and XPS spectra for pristine cathodes before battery operation and the cathodes cycled in various electrolytes. After discharge, the formation of Li₂CO₃ was confirmed by the newly emerged peaks at 860 cm⁻¹, 1412 cm⁻¹ and 1473 cm⁻¹ in the FTIR spectra (**Fig. S16**)²⁷, a peak located at 1089 cm⁻¹ in the Raman spectra (**Fig. S17**)²⁸, and a peak at 290.6 eV in the XPS spectra (**Fig.S18**)^{29,30}. After 3 cycles, the complete Li₂CO₃ decomposition could only be observed in LiFSI cells, based on the disappearance after cycling of the Li₂CO₃ peak in FTIR (**Fig.S16**) and Raman spectra (**Fig. S22**), which indicates outstanding electrochemical reversibility of Li₂CO₃. In contrast, certain amounts of Li₂CO₃ residuals were still detected on the cycled cathodes in the other electrolytes. Examination of the C 1s XPS spectra further revealed the relationship between Li₂CO₃ residuals and battery cycling performance, in which the cells show the presence of Li₂CO₃ residuals except for the cell in LiFSI electrolytes (**Fig. 3c** left).

Furthermore, we have added the XANES spectrum for the pristine cathode in **Fig. 3e** and relevant discussions in the the revised manuscript. As shown in **Fig. 3e**, the C K-edge XANES spectra for the LiNO₃ and LiBF₄ cells displayed the strong peak of Li₂CO₃ at 290.2 eV (* symbol) on cathodes after cycling.³⁶ In addition, the graphite π* (C=C) transition on all cycled cathodes was observed with a slight shift to a higher photon energy compared to the pristine one, indicating an interaction between Li₂CO₃ residuals and rGO catalysts at surfaces. With the existence of C-N species, the cathode cycled in LiFSI-based electrolytes (red line) shows less shift of π* (C=C) peak and much reduced intensity of π*(C=O) peak than that in LiNO₃ cells (yellow line, without C-N species), indicating the weaker interaction between Li₂CO₃ residuals and rGO catalysts, and possible strong interaction between Li₂CO₃ and C-N species,³⁷ in good agreement with our calculation results. It should be noted that the intensities of Li₂CO₃ residuals gradually decreased with increased peak intensity of C-N species at 404.8 eV in the N K-edge XANES spectra³⁸, further confirming the catalytic effects of C-N species towards Li₂CO₃ decomposition.

Supplementary Fig. 16. Fourier transform infrared (FTIR) spectra of the pristine cathode, the discharged cathodes, and the cathodes after 3 cycles in the cells using 1 M LiNO₃, LiBF₄, LiTFSI, and LiFSI electrolytes.

Supplementary Fig. 17. Raman spectra of the cathodes before and after discharge, which were limited to a specific capacity of 500 mA h g^{-1} .

Supplementary Fig. 18. C 1s XPS spectra of the pristine cathode and the cathodes discharged in the cells using 1 M LiNO₃, LiBF₄, LiTFSI, and LiFSI electrolytes. (The peaks at 284.5 eV and 285 eV can be ascribed to non-oxygenated ring carbon (C-C) and the presence of oxygen in rGO (C-O), respectively; C-O/C=O: 287.7 eV, CF₂: ~293 eV).

Supplementary Fig. 22. Raman spectra of the cathodes after 3 cycles in the cells using 1 M LiNO₃, LiBF₄, LiTFSI, and LiFSI electrolytes.

Fig. 3 Characterization of discharge products and SEI composition on the cathodes. (c) C 1s, F 1s, and N 1s XPS spectra for the cathodes after 3 cycles (Li₂CO₃: 290.6 eV, LiF: 685.5 eV, NO₂⁻: 403.8 eV, and NO₃⁻: 408.1 eV, respectively; the bonding configuration of pyrrolic-N at ~400.1 eV is in accord with C-N species).³⁹ (d) The compounds assigned to the C, N, F, and S elements and their relative amounts. (e) C K-edge and N K-edge XANES spectra for the pristine cathode and cathodes after 3 cycles.

Comment 5: In Fig. 4a, the width of the yellow peak seems to be too large.

Reply: Thank you for your insightful comment. We have re-analysed and corrected the peaks of the 1 M LiNO₃/DMSO electrolyte in the Raman spectrum, which can be deconvoluted into free anion (1039 cm⁻¹) and solvent-contact ion pairs (CIP, 1051 cm⁻¹). The relevant description has been amended in the revised manuscript.

It has been widely accepted that a shift of an anion peak to higher wavenumbers in electrolytes indicates that the anion binds with Li⁺ ions, and more shift suggests stronger Li⁺-anion interaction.⁴¹ With increasing shift, the interaction can be classified into free anion, solvent-separated ion pairs (SSIP), and contact ion pairs (CIP).⁴² As shown in Fig. 4a, LiNO₃ and LiFSI electrolytes show more CIP (52.5% and 44.65%, respectively) relative to the other electrolytes, while a smaller proportion of CIP is observed in the others: LiTFSI (13.7%) and LiBF₄ (12.4%) (Tables S2, 3). This demonstrates that NO₃⁻ and FSI⁻ have the strongest coordination with Li⁺ ions compared to the others, which can promote anion-derived SEI generation on electrodes.

Fig. 4 (a) Fitted Raman curves of anions for different anion pairs of free anions, SSIP, and CIP, and their ratio comparison.

Supplementary Table 3. Fitting results for Raman spectra.

Samples	Content in solvent (%)		Content of anions (%)		
	Free	Solvation	Free	SSIP	CIP
1M LiNO ₃	57.75	42.25	42.5	0	52.5
1M LiBF ₄	53.25	46.75	0	87.6	12.4
1M LiTFSI	50.31	49.96	35.1	51.2	13.7
1M LiFSI	52.7	47.3	0	55.35	44.65

Comment 6: Are there some experimental proofs for the formation of Li–N bonds?

Reply: Thank you for your kind comment. Based on our calculated results (**Fig.1d**), the atomic configurations clearly show that there is a strong interaction between Li₂CO₃ and C–N species, revealing the weaker nature of the Li–O bonds and reduced decomposition energy barriers of Li₂CO₃.

Fig. 1 Top views and side views of the atomic structures and the charge density differences of (**d**) Li₂CO₃ adsorption on graphene, C-F, and C-N species. The red numbers are the Bader charge values of these different species for Li₂CO₃ molecules. The yellow and blue zones represent electron loss and gain, respectively (isovalue, 2×10^{-6}). The carbon, oxygen, fluorine, nitrogen, sulphur, and lithium atoms are marked as copper, red, silver, purple, yellow, and light green, respectively.

However, in this work, the C–N species is covered by Li₂CO₃ on the surface of cathodes, making it difficult to detect the interaction between Li₂CO₃ and C–N species in experiments. Therefore, we have conducted the XANES analysis for cathodes before and after cycling to trace the residuals of Li₂CO₃, rGO as substrate matrix, and C–N species due to its high sensitivity for carbon detection. This may offer a clue of the interaction between Li₂CO₃ and C–N species to some extent via detecting the interaction between Li₂CO₃ residuals and rGO, and the existence of C–N species. In the C K-edge XANES spectrum for the pristine cathode, there are three main peaks of electron transitions at 285.4, 288.6, and 292.7 eV,

respectively, corresponding to the $\pi^*(\text{C}=\text{C})$, $\pi^*(\text{C}=\text{O})$, and $\sigma^*(\text{C}=\text{C})$. Among them, the $\pi^*(\text{C}=\text{O})$ transition comes from the oxygen-containing functional groups of rGO.[1-4] As shown in Fig. 3e, the C K-edge XANES spectra for the LiNO_3 and LiBF_4 cells displayed the strong peak of Li_2CO_3 at 290.2 eV (\star symbol) on cathodes after cycling.³⁶ In addition, the graphite $\pi^*(\text{C}=\text{C})$ transition on all cycled cathodes was observed with a slight shift to a higher photon energy compared to the pristine one, indicating an interaction between Li_2CO_3 residuals and rGO catalysts at surfaces. With the existence of C-N species, the cathode cycled in LiFSI-based electrolytes (red line) shows less shift of $\pi^*(\text{C}=\text{C})$ peak and much reduced intensity of $\pi^*(\text{C}=\text{O})$ peak than that in LiNO_3 cells (yellow line, without C-N species), indicating the weaker interaction between Li_2CO_3 residuals and rGO catalysts, and possible strong interaction between Li_2CO_3 and C-N species,³⁷ in good agreement with our calculation results. It should be noted that the intensities of Li_2CO_3 residuals gradually decreased with increased peak intensity of C-N species at 404.8 eV in the N K-edge XANES spectra³⁸, further confirming the catalytic effects of C-N species towards Li_2CO_3 decomposition. The relevant discussions have been amended in the revised manuscript.

Fig. 3 (e) C K-edge and N K-edge XANES spectra for the pristine cathode and cathodes after 3 cycles.

[1] Kikuma, J.; Tonner, B.P. XANES spectra of a variety of widely used organic polymers at the C K-edge. *J. Electron Spectrosc. Relat. Phenom.* **82**, 53-60 (1996).

[2] Ray, S. C. et al. Orientation of graphitic planes during annealing of “dip deposited” amorphous carbon film: A carbon Kedge X-ray absorption near-edge study. *Carbon* **44**, 1982-1985 (2006).

[3] Zhou, J.G. et al. Nano-scale chemical imaging of a single sheet of reduced graphene oxide. *J. Mater. Chem.* **21**, 14622-14630 (2011).

[4] Zhou, J. et al. Nanoscale chemical imaging and spectroscopy of individual RuO_2 coated carbon nanotubes. *Chem. Commun.* **46**, 2778-2780 (2010).

Comment 7: What's the effect of different C–N configurations on the catalytic performance?

Reply: Thank you for your valuable comment. As shown in **Figure 1**, our calculated results show there are three possible atomic configurations of C-N bonds, corresponding to the pyrrolic-N (denoted as C-N species, ~400.1 eV), pyridinic-N, and graphitic-N. In this work, however, pyrrolic-N (398.6 eV) and graphitic-N (401.4 eV) were not detected on the cycled cathodes based on the N 1s XPS spectra (**Fig.3c**). Therefore, we have not provided relevant discussion about the effect of pyridinic-N and graphitic-N on the catalytic performance in the manuscript.

First-principles calculation results show that C-N species show relatively higher adsorption energies towards CO_2 (0.1 eV) and $^*\text{CO}_2^{2-}$ (-1.04 eV) than those of pyridinic-N and graphitic-N (**Figure 1e**, yellow and red bars, respectively) in the CRR, indicating a lower formation barrier of Li_2CO_3 and higher catalytic activation on C-N species. In the subsequent CER, the adsorption energies show that C-N species also exhibits the strongest interaction (-1.43 eV) with Li_2CO_3 , whereas pyridinic-N, and graphitic-N show weak interactions (-0.79 and -0.41 eV, respectively, green bar). This suggests that Li_2CO_3 on C-N species can more easily lose electrons to decompose than that on pyridinic-N and graphitic-N. As a result, C-N species can provide stronger adsorption sites and fast charge transfer capability to $^*\text{CO}_2^{2-}$ and Li_2CO_3 than other C-N configurations, achieving boosted CRR/CER kinetics and enhanced interfacial catalytic properties.

Figure 1. Top view of the atomic configurations of (a) pyrrolic-N, (b) pyridinic-N, and (c) graphitic-N. The carbon, oxygen, and nitrogen atoms are marked as copper, red, and purple, respectively. (d) Schematic illustration of nitrogen species with the corresponding reported XPS binding energies in the N-doped graphene.[1] (e) Adsorption energies of graphene, C-N (pyrrolic-N, pyridinic-N, and graphitic-N), C-F, C-O, and C-S species for CO_2 , $^*\text{CO}_2^{2-}$, and Li_2CO_3 .

[1] He, H. et al. Investigation of edge-selectively nitrogen-doped metal free graphene for oxygen reduction reaction. *J. Adv. Nanotechnol.* **1**, 5-13 (2020).

Fig. 3 (c) C 1s, F 1s, and N 1s XPS spectra for the cathodes after 3 cycles (Li_2CO_3 : 290.6 eV, LiF: 685.5 eV, NO_2^- : 403.8 eV, and NO_3^- : 408.1 eV, respectively; the bonding configuration of pyrrolic-N at ~400.1 eV is in accord with C-N species).

REVIEWER COMMENTS

Reviewer #1 (Remarks to the Author):

The revised manuscript has shown substantial improvements. Most of my concerns have been addressed. The manuscript could be recommended for publication. Meanwhile, there are a few more points to be corrected.

1. In the authors' response, they argued that the Li anode cannot be replaced by other anode materials, such as LFP/LTO, in Li-CO₂ batteries. However, I cannot agree with this. These "pseudo" anodes could also release Li⁺ to react with CO₂ and form discharge product. Stable pseudo anodes have been used frequently in literature to eliminate the potential influence of the unstable Li anode. Anyway, the test with replaced fresh Li anode is an alternative support.
2. Since the existence and the role of the C-N species is the core point for this work, it is recommended to add infrared data to further prove this.
3. The assignment of the Raman data of LiFSI electrolyte in Fig.4a is not clear. What did the peaks between 630-720 cm⁻¹ stand for?

Reviewer #2 (Remarks to the Author):

Upon revision with additional experimental evidence, the manuscript is dramatically improved. I do not have any further questions.

Reviewer #3 (Remarks to the Author):

The authors have addressed most issues raised by the reviewers and the revised version can thus be accepted for publication.

Point-by-Point Response to the Reviewers' Comments

Reply to Reviewer #1

Overall comment: The revised manuscript has shown substantial improvements. Most of my concerns have been addressed. The manuscript could be recommended for publication. Meanwhile, there are a few more points to be corrected.

Reply: Thank you very much for your insightful comments and positive opinion on our manuscript. We have carefully revised the manuscript according to your suggestions.

Comment 1: In the authors' response, they argued that the Li anode cannot be replaced by other anode materials, such as LFP/LTO, in Li-CO₂ batteries. However, I cannot agree with this. These "pseudo" anodes could also release Li⁺ to react with CO₂ and form discharge product. Stable pseudo anodes have been used frequently in literature to eliminate the potential influence of the unstable Li anode. Anyway, the test with replaced fresh Li anode is an alternative support.

Reply: Thank you for your valuable suggestion. We agree that "pseudo" anodes can release Li⁺ ions to react with CO₂ and form discharge products. Lithium titanate (Li₄Ti₅O₁₂, LTO) has been employed to replace lithium metal because of its flat working potential at around 1.55 V (vs. Li/Li⁺), as in your kind comments .[1-3]

To evaluate the electrochemical performance of LTO in LTO-CO₂ batteries, LTO was first discharged in Li/LTO half-cells to insert active Li⁺, and then paired with air cathodes (carbon papers loading rGO as catalyst) to assemble LTO-CO₂ full-cells. **Figure 1a** shows the cycling performance of LTO-CO₂ cells using 1 M LiTFSI and 0.25 M LiNO₃/0.75 M LiFSI electrolytes at the current density of 0.1 A g⁻¹ over the voltage window of 0-4.5 V. These cells exhibit dramatically low discharge voltages (just above 0.5 V) and poor capacities (less than 10 mAh g⁻¹). It should be noted that the cells using the optimized 0.25 M LiNO₃/0.75 M LiFSI electrolyte performed a relatively larger capacity and stable discharge/charge voltage plateau compared to that using the LiTFSI/DMSO electrolyte (**Figure 1b**). This demonstrates the advantages of our electrolytes towards improving electrochemical performance in CO₂-based batteries. The theoretical specific capacity of LTO is 175 mAh g⁻¹[4,5], which is significantly lower than that (3,860 mAh g⁻¹) of lithium metal.[6,7] This may cause the lack of active Li⁺ ions for the formation of Li₂CO₃, thus resulting in the notably reduced capacity in LTO-CO₂ batteries. Thus, searching for an alternative anode material with a high theoretical capacity and superior stability to replace lithium metal anodes might be a key challenge for CO₂-based batteries in the future.

Figure 1. (a) Long-term cycling performance of LTO- CO_2 cells at 0.1 A g^{-1} over the voltage window of 0-4.5 V. (b) Discharge-charge profiles at 10th cycle of Li/LTO half-cells using the 1 M LiTFSI and 0.25 M $\text{LiNO}_3/0.75 \text{ M LiFSI}$ electrolytes.

[1] Park, K.S., Benayad, A., Kang, D.J. and Doo, S.G., 2008. Nitridation-driven conductive $\text{Li}_4\text{Ti}_5\text{O}_{12}$ for lithium ion batteries. *J. Am. Chem. Soc.* **130**, 14930-14931 (2008).

[2] Lin, C. et al. $\text{Li}_4\text{Ti}_5\text{O}_{12}$ -based anode materials with low working potentials, high rate capabilities and high cyclability for high-power lithium-ion batteries: a synergistic effect of doping, incorporating a conductive phase and reducing the particle size. *J. Mater. Chem. A* **2**, 9982-9993 (2014).

[3] Chen, S. et al. Self-supported $\text{Li}_4\text{Ti}_5\text{O}_{12}$ nanosheet arrays for lithium ion batteries with excellent rate capability and ultralong cycle life. *Energy Environ. Sci.* **7**, 1924-1930 (2014).

[4] Wagemaker, M. et al. A kinetic two-phase and equilibrium solid solution in spinel $\text{Li}_{4+x}\text{Ti}_5\text{O}_{12}$. *Adv. Mater.* **18**, 3169-3173 (2006).

[5] Chiu, H.C. et al. Capacity fade mechanism of $\text{Li}_4\text{Ti}_5\text{O}_{12}$ nanosheet anode. *Adv. Energy Mater.* **7**, 1601825 (2017).

[6] Jiao, S. et al. Behavior of lithium metal anodes under various capacity utilization and high current density in lithium metal batteries. *Joule* **2**, 110-124 (2018).

[7] Fang, C. et al. Quantifying inactive lithium in lithium metal batteries. *Nature* **572**, 511-515 (2019).

Comment 2: Since the existence and the role of the C-N species is the core point for this work, it is recommended to add infrared data to further prove this.

Reply: Thank you for your insightful suggestion. We have provided FTIR spectra (Fig. S16) for the cathodes cycled in various electrolytes to prove the catalytic role of C-N species towards the decomposition of Li_2CO_3 .

Supplementary Fig. 16. Fourier transform infrared (FTIR) spectra of the pristine cathode, the discharged cathodes, and the cathodes after 3 cycles in the cells using 1 M LiNO₃, LiBF₄, LiTFSI, and LiFSI electrolytes. An enlargement for the cycled cathodes in the range of 1300-1600 cm⁻¹ is shown on the right.

Note: After 3 cycles, the peaks at around 1340 and 1510 cm⁻¹ (Fig. S16) are assigned to in-plane bending and stretching vibrations of C-N species, respectively,¹⁻³ confirming the existence of C-N species formed on the surfaces of the cathodes. The FTIR spectra show the highest peak intensities of C-N species on the cathode cycled in the LiFSI-based electrolytes compared to those in the other electrolytes, consistent with the results of no Li₂CO₃ residuals observed in LiFSI-based electrolytes. In contrast, certain amounts of Li₂CO₃ residuals were detected in the LiNO₃ and LiTFSI electrolytes with lower intensities of C-N species. These observations identified a positive correlation between C-N species and the reversibility of Li₂CO₃, consistent with the XPS and XANES results in this work.

1. Tetsuka, H. et al. Molecularly designed, nitrogen-functionalized graphene quantum dots for optoelectronic devices. *Adv. Mater.* **28**, 4632-4638 (2016).
2. Permatasari, F.A. et al. Design of pyrrolic-N-rich carbon dots with absorption in the first near-infrared window for photothermal therapy. *ACS Appl. Nano Mater.* **1**, 2368-2375 (2018).
3. Yang, J. et al. Adsorption-catalysis synergy within pyrrolic-N-rich carbon nanosheets: Propelling electrochemical kinetics and shielding polysulfides for lithium-sulfur batteries. *Chem. Eng. J.* **476**, 146532 (2023).

Comment 3: The assignment of the Raman data of LiFSI electrolyte in Fig.4a is not clear. What did the peaks between 630-720 cm⁻¹ stand for?

Reply: Thank you for your kind comment. We have labelled the peaks between 630-720 cm⁻¹ in the

Raman spectrum of LiFSI electrolyte in **Fig.4a** and added the relevant description in the revised manuscript.

Part of Fig. 4 (a) Fitted Raman curves of FSI⁻ anions in the range of 630-720 cm⁻¹ for different anion pairs of free anions, SSIP, and CIP. The peaks located at 667 and 697 cm⁻¹ can be assigned to the C-S symmetric and asymmetric stretching vibrations of free DMSO, respectively. The peaks at 676 and 708 cm⁻¹ are attributed to the C-S symmetric and asymmetric stretching vibrations of DMSO molecules that solvate with Li⁺ ions.^{17,40-42}

17. Zhang, W. et al. Regulating the reduction reaction pathways via manipulating the solvation shell and donor number of the solvent in Li-CO₂ chemistry. *Proc. Nat. Acad. Sci.* **120**, 2219692120 (2023).

40. He, M. et al. Concentrated electrolyte for the sodium-oxygen battery: solvation structure and improved cycle life. *Angew. Chem. Int. Ed.* **55**, 15310-15314 (2016).

41. Yamada, Y. et al. Electrochemical lithium intercalation into graphite in dimethyl sulfoxide-based electrolytes: effect of solvation structure of lithium ion. *J. Phys. Chem. C* **114**, 11680-11685 (2010).

42. Wang, L., Uosaki, K. and Noguchi, H. Effect of electrolyte concentration on the solvation structure of gold/LiTFSI-DMSO solution interface. *J. Phys. Chem. C* **124**, 12381-12389 (2020).

REVIEWERS' COMMENTS

Reviewer #1 (Remarks to the Author):

The reviewer's questions have been addressed. The manuscript is recommended for publication.